

# Moisture transport axes: a unifying definition for monsoon air streams, atmospheric rivers, and warm moist intrusions

Clemens Spensberger[1], Kjersti Konstali[1], and Thomas Spengler[1]

[1]Geophysical Institute, University of Bergen, and Bjerknes Centre for Climate Research, Bergen, Norway

**Correspondence:** Clemens Spensberger (clemens.spensberger@uib.no)

**Abstract.** The water vapor transport in the extratropics is mainly organized in narrow elongated filaments. These filaments are referred to with a variety of names depending on the contexts, for example atmospheric river, warm moist intrusion, warm conveyor belt, and feeder air stream. Despite the various names, these features share essential properties, such as their narrow elongated structure. Here, we propose an algorithm that detects these various lines of moisture transport in instantaneous maps of the vertically integrated water vapor transport. The detection algorithm extracts well-defined maxima in the water vapor transport and connects them to lines that we refer to as moisture transport axes. By only requiring a well-defined maximum in the vapor transport, we avoid imposing a threshold in the absolute magnitude of this transport or the total column water vapor. Consequently, the algorithm is able to pick up moisture transport axes at all latitudes without requiring region-specific tuning or normalization. We demonstrate that the algorithm can detect both atmospheric rivers and warm moist intrusions, but also prominent monsoon air streams as well as low-level jets with moisture transport. Atmospheric rivers sometimes consist of several distinct moisture transport axes, indicating the merging of several moisture filaments into one atmospheric river. We showcase the synoptic situations and precipitation patterns associated with the occurrence of the identified moisture transport axes in example regions in the low, mid, and high latitudes.

## 1 Introduction

Throughout the mid-latitudes, the bulk of the moisture transport occurs in narrow filaments (Newell et al., 1992; Zhu and Newell, 1994, 1998; Sodemann and Stohl, 2013; Dacre et al., 2015) that received much attention under the label *atmospheric river* (e.g., Zhu and Newell, 1994; Ralph et al., 2005; Lora et al., 2020). Landfalling atmospheric rivers are key contributors to the hydrological cycle and can produce intense precipitation events that lead to flooding (e.g., Zhu and Newell, 1994; Sodemann and Stohl, 2013). Dynamically, atmospheric rivers are generally linked to one or more extratropical cyclones, with anticyclones often contributing to the filamentation of the moisture field (Azad and Sorteberg, 2017; Zhang et al., 2019). Conversely, the moisture contained in atmospheric rivers can contribute to cyclone intensification through the feeder airstream (Dacre et al., 2019).

Recent intercomparisons based on reanalyses and climate projections, however, highlight that the definitions of atmospheric rivers diverge widely (Atmospheric River Tracking Method Intercomparision Project; ARTMIP; Rutz et al., 2019; Collow et al., 2022; O'Brien et al., 2022; Shields et al., 2023). This might be a reflection of the fact that atmospheric rivers can form in



relation to different meteorological phenomena (Gimeno et al., 2021). Nevertheless, given the large variety in definitions, the choice of the diagnostic variable, threshold, and normalization method remains a subjective choice that often depends on the study region. Alternatively, these choices can be left to a machine-learning algorithm (O'Brien et al., 2020). We introduce an alternative approach that focuses on the structure of moisture filaments rather than their intensity.

Moisture filaments similar to atmospheric rivers also occur at higher latitudes (Sorteberg and Walsh, 2008; Woods and Caballero, 2016; Papritz et al., 2022). In the Arctic, such features would typically be called *warm moist intrusions* (Woods and Caballero, 2016; Papritz et al., 2022). Gorodetskaya et al. (2014) and Wille et al. (2019, 2021), however, apply the same concept and label moisture filaments making landfall in Antarctica as *atmospheric rivers*. Analogously, Mattingly et al. (2018) use the concept to study moisture transport onto the Greenland Ice Sheet. Although using the same label, atmospheric rivers

in polar regions either require region-specific tuning or normalization of the input fields to accommodate for the much lower absolute moisture content compared to, for example, California (cf. differences in polar and mid-latitude-focused algorithms in ARTMIP; Shields et al., 2022).

    In the subtropics, the concept *atmospheric river* has only been sporadically used, potentially due to reduced day-to-day variability of weather in this region (e.g., Vallis, 2006; Ogawa and Spengler, 2019). Still, Viale and Nuñez (2011) and Dezfuli

(2020) link precipitation in subtropical South America and the Middle East to this concept, but focus on regions that are still intermittently affected by mid-latitude weather systems. Park et al. (2021) use atmospheric rivers to conceptualize month-to-month variability in the East Asian Monsoon and thus apply the concept to a truly subtropical weather pattern. In addition, topographically steered moisture transport in the subtropics, such as along the South American low level jet, are sometimes considered to constitute atmospheric rivers (e.g., Poveda et al., 2014). Despite these examples, it remains unclear to what extent

the concept can and should be extended to the subtropics.

    We therefore aim to more generally detect moisture filaments based on their elongated structure. We adopt this approach from an upper tropospheric jet detection algorithm where jets are defined as lines of maximum winds at the dynamical tropopause (Spensberger et al., 2017; Spensberger and Spengler, 2020). We apply this approach to the vertically integrated water vapor transport (IVT) and trace lines of maximum moisture transport. We do not require a minimum threshold of IVT, but rather that

the maximum in the moisture transport is well-defined. We call the resulting lines *moisture transport axes*.

    This approach is related to the atmospheric river definitions in Ullrich and Zarzycki (2017); Ullrich et al. (2021) and Nel-likkattil et al. (2024), who use combinations of second derivatives of the IVT field as the basis for the detection. While motivated by the same need for a more structure-based definition, their threshold fields still scale linearly with IVT magnitude and thus provide only limited advantage over using IVT-thresholds directly.

Our approach is most similar to atmospheric river detection algorithms that include transport axes to define the start, end, or length of an atmospheric river. Mundhenk et al. (2016) and Inda-Díaz et al. (2021) derive such an axis a-posteriori from the two-dimensional atmospheric river object, thus relying on geometrical properties that do not have immediate physical significance. Similarly, Wick et al. (2013) and Xu et al. (2020) derive their axes by applying techniques from image processing. Lavers et al. (2012) and Griffith et al. (2020), like us, define the axis by the maximum transport, but require a target region and consider

only lines of maxima that continuously extend westward from that region. Finally, and most similar to our approach, Guan



and Waliser (2015) define their axis such that it highlights the maximum IVT within their two-dimensional detections. Their construction method, however, cannot detect overturning axes, which can occur close to the core of extratropical cyclones, and only provides a sometimes incoherent set of axis grid points rather than a continuous line. In contrast to previous approaches, our algorithm is directly and only based on the structure of the IVT vector field and identifies elongated maxima in the
IVT irrespective of their orientation. Geometric features like start point, end point, and length are thus straightforward and unambiguously defined.

There are several further advantages of considering moisture filaments as a one-dimensional rather than two-dimensional feature. First, as we will show, an atmospheric river outline can contain several distinct maxima in the moisture transport where these maxima can even be oriented in nearly opposing directions. Finally, moisture transport axes visually highlight the
direction of the moisture transport, for example relative to the orientation of a coastline, which is essential to assess orographic precipitation (Griffith et al., 2020).

## 2   Data and detection method

We base our study on 3-hourly ERA5 reanalysis data at $0.5°$ resolution for the period 1979-2020 (Hersbach et al., 2020). We follow Spensberger et al. (2017) and Spensberger and Spengler (2020) and spectrally filter the IVT components to T84
resolution as the detection algorithm is somewhat sensitive to grid point noise. The grid remains unchanged by the filtering. This resolution is fine enough to retain all synoptic scale and many mesoscale structures (cf., Spensberger et al., 2017, and the case study snapshots herein; Figs. 2; 5a,b; 6a,b; 7a,b; and S1 in the supplement). The IVT components are used as provided by the European Centre for Medium-Range Weather Forecasts (ECMWF).

The detection identifies lines of maximum IVT (method illustrated in Fig. 1), analogous to the jet axis detection tracing lines
of maximum wind (Spensberger et al., 2017). The algorithm identifies well-defined maxima in IVT in cross sections normal to the IVT direction (e.g., the sections in Fig. 1). In practice, we find locations where the "shear" $\sigma$ in IVT,

$$\sigma_{\text{IVT}} = \frac{\partial |\text{IVT}|}{\partial n} = 0 \quad , \tag{1}$$

by checking for all pairs of neighboring grid points whether $\sigma_{\text{IVT}}$ has changed sign. Here, $n$ is the direction perpendicular to the direction of the local water vapor transport. This procedure is analogous to the jet axis detection which identifies the location
of the wind maximum by the zero-shear line, the line marking the transition from cyclonic to anticyclonic shear. We pinpoint the exact location of the $\sigma_{\text{IVT}} = 0$-line by linear interpolation between identified pairs of neighboring grid points.

In the second step, we filter out IVT minima and weak IVT maxima by requiring

$$|\text{IVT}| \cdot \frac{\partial \sigma_{\text{IVT}}}{\partial n} \leq K_{\text{IVT}} \quad , \tag{2}$$

where $K_{\text{IVT}}$ is a (negative) threshold to be determined. This criterion extracts well-defined maxima in the IVT using a combi-
nation of absolute magnitude ($|\text{IVT}|$) and sharpness of the IVT maximum ($\frac{\partial \sigma_{\text{IVT}}}{\partial n}$). Small-amplitude maxima can thus become part of a moisture transport axis if the associated peak in IVT is sharp enough (yellow dots in the cross sections A and B in





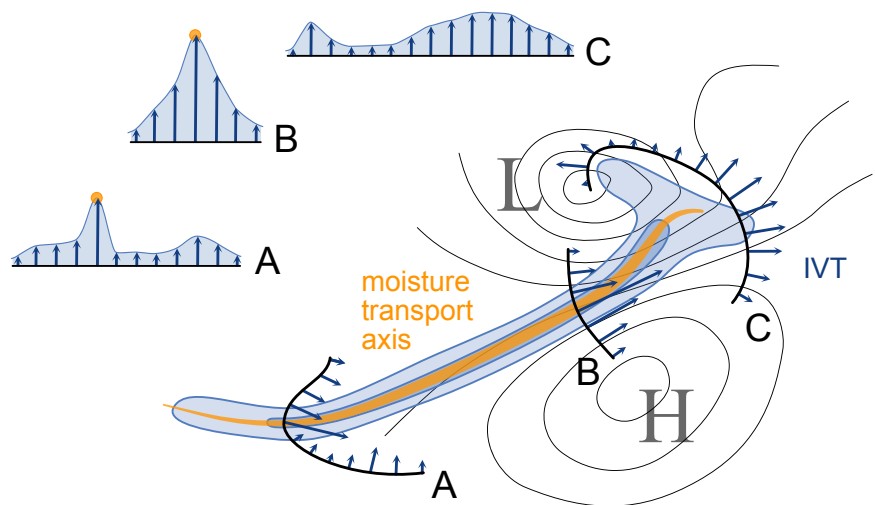

**Figure 1.** Illustration of the detection method. Blue contours and vectors show the magnitude and direction of the vertically integrated water vapor transport (IVT). The black lines marked A, B and C are cross sections perpendicular to the local IVT direction. Well-defined maxima in the local IVT along the sections are marked by yellow circles in A and B; these well-defined maxima are then connected to form the yellow moisture transport axis line. Black contours indicate isobars, and the labels L and H mark a cyclone and an anticyclone, respectively.

Fig. 1). Minima in IVT do not fulfill this criterion, because there $\frac{\sigma_{\mathrm{IVT}}}{\partial n} > 0$. For determining $K_{\mathrm{IVT}}$, we follow the suggestion of Spensberger et al. (2017) and use the 12.5-percentile of year-round global $|\mathrm{IVT}| \cdot \frac{\partial \sigma_{\mathrm{IVT}}}{\partial n}$. For ERA5 the resulting threshold is $K_{\mathrm{IVT}} = -4.06 \cdot 10^{-7}\,\mathrm{kg}^2\mathrm{s}^{-2}\mathrm{m}^{-4}$. When applied to a climate model, the variation of this percentile across the climates to be studied will provide an indication on whether the threshold might need to be adapted. However, given that the definition of the axes is based on their elongated structure and that the algorithm works well across different current climate zones (see following sections), the threshold might not need to be adapted across simulations for typical climate change scenarios.

In a third step, we connect all remaining points marking well-defined IVT maxima into lines using a maximum distance of 1.5 grid points between two successive points along a line (technical details on this step in Spensberger et al., 2017). In the fourth and final step, we require a minimum length of 2000 km, measured following the line, for such a line to become a moisture transport axes (yellow line in Fig. 1). This minimum length is in line with typical geometry constraints used for detecting atmospheric rivers (Rutz et al., 2019).

In all following snapshots, the resulting moisture transport axes are directly visualized as yellow-white lines. For visualizing time mean occurrence in climatologies and composites, we however find one additional step to be helpful. As lines do not have an areal extent, appropriate normalization without further assumptions would result in the unit $\mathrm{m}^{-1}$ for mean occurrence. This unit can be interpreted as an average line length per unit area (e.g., Spensberger and Spengler, 2020) but is still not intuitive. Thus, we use a 200 km-radius to lend our lines an areal extent. A small radius such as that does hardly change the geographical





pattern, but the time-mean occurrence of this spatial feature becomes simply a dimensionless fraction of time steps in which a moisture transport axis is within 200 km of a given location.

## 3    Case studies from low, mid, and high latitudes

To illustrate the performance of the detection algorithm, we first showcase selected occurrences of moisture transport axes (yellow lines in Fig. 2). Three of the four cases in Fig. 2 are based on previous studies of atmospheric rivers. The North Pacific and South Indian Ocean cases are discussed in Lora et al. (2020) (our Fig. 2a-d). The North Atlantic case (Fig. 2e,f) is discussed in Azad and Sorteberg (2017) and yielded one of the highest daily precipitation totals on record in Bergen on the west coast of Norway. These cases also include examples of high-latitude moisture transport axes, such as the one touching the Antarctic coastline south of Africa (Fig. 2d) and the one close to Novaya Zemlya (Fig. 2f). The fourth case highlights a moisture transport axis from the Sahel region across the Sahara towards the Mediterranean (Fig. 2g,h). This moisture transport axis is associated with scattered patches of precipitation exceeding 1 mm/h over the Sahara desert (magenta contours in Fig. 2g,h).

At all latitudes, the showcased moisture transport axes clearly trace maxima in the moisture transport (right column of Fig. 2), despite the large variation in the absolute magnitude of the IVT along the various axes and across the different climate zones. Many axes extend to IVT values below $250 \, \mathrm{kg\,s^{-1}\,m^{-1}}$ and some trace IVT maxima beyond the saturation of the color scale at $1150 \, \mathrm{kg\,s^{-1}\,m^{-1}}$.

For mid-latitude moisture transport axes, the cases suggest a qualitatively good correspondence to typical atmospheric river detections. More specifically, our moisture transport axes correspond well to the six atmospheric river detection schemes available for ERA5 [red, orange and pale contours in the right column of Fig. 2; cf. also majority consensus detections for the cases in Fig. 4c,g of Lora et al. (2020)]. In contrast to atmospheric rivers, however, moisture transport axes highlight the diffluent moisture transport close to the occlusion point of a cyclone (e.g., Fig. 2b). From the occlusion point, one branch of moisture transport typically spirals cyclonically towards the cyclone core, while another branch follows the warm front away from the cyclone core. The dominant branch varies from cyclone to cyclone (towards cyclone in Fig. 2b, away in Fig. 2d, both branches evident in Fig. 2f).

Beyond the mid-latitudes, moisture transport axes highlight moisture transport that might not generally be considered an atmospheric river and that is less consistently identified as such by other schemes. For example, the moisture transport axes pick up a moisture filament associated with only moderate IVT close to Novaya Zemlya that is picked up by one to three of the six global detection algorithms in the ARTMIP ERA5 catalogue (outer contour: GuanWaliser_v2; inner contour: Mundhenk_v3 and ClimateNet; Fig. 2e,f). Moisture transport axes also pick up the cyclonic and nearly circular moisture transport around tropical cyclones (e.g., close to the North American east coast in Fig. 2f), as well as moderate-intensity moisture transport filaments over North Africa (Fig. 2f). Further, the moisture transport axes trace an expulsion of Tropical moisture into the much drier subtropics which is classified as an atmospheric river by two of the six available algorithms (outer contour: GuanWaliser_v2; large parts also detected by Mundhenk_v3 [not shown]; Fig. 2h). A more thorough comparison with atmospheric rivers follows in section 5.



## 4 Climatology of moisture transport axes

Climatologically, the occurrence of moisture transport axes in the mid-latitudes follows the storm tracks [Fig. 3; cf. Chang et al. (2002) and Wirth et al. (2018)]. This is particularly true for the winter hemisphere (Fig. 3a,c). During winter, the occurrence of moisture transport axes is closely related to the IVT climatology (gray contours in Fig. 3), whereas during summer and autumn, moisture transport axes occur frequently over the continents downstream of a storm track despite comparatively small climatological IVT.

In the subtropics and tropics, the occurrence of moisture transport axes is confined to specific regions (Fig. 3). Moisture transport axes frequently occur along the Intertropical Convergence Zone (ITCZ) year-round. In addition, the monsoon circulation is frequently associated with moisture transport axes, for example the Indian monsoon (Fig. 3c) as well as the Somali Jet (Moisture transport in the Somali Jet discussed in detail in Viste and Sorteberg, 2013). The climatology of moisture transport axes also highlights the moisture transport along low-level jets steered by orography. This effect is apparent year-round along the South American low-level jet on the eastern side of the Andes (Montini et al., 2019), but also, with seasonal variation, along the straits through the Maritime continent.

In comparison with the low and mid-latitudes, relatively few moisture transport axes occur in subpolar and polar regions (Fig. 3). Nevertheless, the 1%-frequency of occurrence isoline extends to around 80°N in the North Atlantic during winter (Fig. 3a) and the 5%-contour covers most of the ice-free Arctic Ocean during summer (Fig. 3c). Similarly, in the North Pacific, the Aleutian Islands have a moisture transport axis nearby for more than 5% of the time winter time steps. In the Southern Hemisphere, the 1%-isoline generally follows the Antarctic coastline during all seasons and extends furthest poleward upstream of the Antarctic peninsula in the South Pacific sector. Over the ice sheets of Antarctic and Greenland, very few moisture transport axes are detected.

In synthesis, the case studies and climatologies give a strong indication that our definition of moisture transport axes is able to detect moisture filaments across all but the driest climate zones. The detections appears to align well with the concept of atmospheric rivers in the mid-latitudes (a more stringent comparison follows in sec. 5), but also capture synoptically meaningful events in the tropics, subtropics, and polar regions.

## 5 Relation to atmospheric rivers

The climatologies of moisture transport axes resemble the consensus detection frequencies in the ARTMIP catalog (compare our Fig. 3 with Fig. 5 of Collow et al., 2022). The most obvious difference between our results and the ARTMIP consensus is the frequency of detection in the subtropics and tropics. We observe a significant number of detections at low latitudes. These might not be desirable for all applications and we thus explore the potential of reducing the number of tropical detections by normalizing the IVT input fields in the supplement. In the mid-latitudes, the results are however largely unaffected by normalization (cf. supplement).

To corroborate the qualitatively good correspondence between moisture transport axes and atmospheric rivers documented by the case studies and climatologies, we supplement these analyses first by a quantitative comparison of the properties of mois-



ture transport axes with a selection of criteria that are sometimes used to identify atmospheric rivers (cf. marginal histograms

in Fig. 4), and second by a composite analysis of moisture transport axes making landfall in Northern California.

The distribution of MTA occurrence across latitude is clearly bimodal with one narrow peak around 10° latitude as well as a wider peak centered around 45° latitude (Fig. 4a-d). The minimum in frequency of detections occurs around 20° latitude, close to the 15° and 23.25°-cutoffs used in the atmospheric river detection schemes of Ullrich and Zarzycki (2017) and Shearer et al. (2020), respectively.

The bimodality in latitude for the non-normalized detections is reflected in a slight bimodality also in total column water vapor (TCWV; Fig. 4a). Tropical moisture transport axes are associated with distinctly more TCWV than extratropical moisture transport axes, with only a small overlap in the distributions around $45\,\mathrm{kg\,m^{-2}}$. The most frequent TCWV in extratropical axes is approximately $20\,\mathrm{kg\,m^{-2}}$ and around 20% of the moisture transport axes are below this value. Note that the histogram is based on all points along all moisture transport axes rather than the axes' peak intensity. The statistic thus implies that 20% of

the combined length of all moisture transport axes is below the $20\,\mathrm{kg\,m^{-2}}$ threshold.

Most detection algorithms for atmospheric rivers use IVT as one of their criteria to define the feature. Typical thresholds used are 250, 500 and $700\,\mathrm{kg\,s^{-1}\,m^{-1}}$ (e.g., CONNECT by Sellars et al. (2015), TempestExtremes by Ullrich and Zarzycki (2017), and the Rutz et al. (2014) algorithm). In comparison, the most common IVT value along moisture transport axes is around $350\,\mathrm{kg\,s^{-1}\,m^{-1}}$ (Fig. 4b). Interestingly, the most common value is largely independent of latitude (Fig. 4b).

In particular, typical tropical and extratropical moisture axes are associated with similar IVT. The strongest IVT, beyond $700\text{-}800\,\mathrm{kg\,s^{-1}\,m^{-1}}$, does however mostly occur around 45° latitude. Consequently, the vast majority (89%) of the transport axes exceed the $250\,\mathrm{kg\,s^{-1}\,m^{-1}}$-threshold. At the same time, most of the axes are also located below the $500\,\mathrm{kg\,s^{-1}\,m^{-1}}$ and $700\,\mathrm{kg\,s^{-1}\,m^{-1}}$-thresholds (66 and 88%, respectively).

Instead of using a cut-off latitude, several definitions of atmospheric rivers rely on a threshold for the poleward and/or

eastward component of the IVT (e.g., Guan and Waliser, 2015; Mattingly et al., 2018; Viale et al., 2018). The eastward component of the IVT also exhibits a bimodal distribution with a local minimum between the westerly and easterly modes at $0\,\mathrm{kg\,s^{-1}\,m^{-1}}$ (Fig. 4c). This bimodality maps reasonably well onto the bimodality in latitude, with tropical axes being mostly easterly and mid-latitude axes being mostly westerly. A separation using a threshold at zero nevertheless remains questionable, as there is still a substantial number of mid-latitude axes with easterly moisture transport. This happens most frequently around

60° latitude (Fig. 4c). From a synoptic perspective, this is most often associated with the easterlies on the poleward side of a cyclone (example in Fig. 2b). A further problem with a cut-off at $0\,\mathrm{kg\,s^{-1}\,m^{-1}}$ eastward IVT is the frequent occurrence of mainly meridional moisture transport in the Southern Hemisphere (e.g., Fig. 2d), where slight variations off the meridional direction then determine whether an atmospheric river is detected.

Similar arguments hold for the poleward IVT (Fig. 4d). Here, the distribution is unimodal and centered close to zero poleward

IVT. Consequently, almost as many axes exhibit equatorward and poleward moisture transport (equatorward transport along 44% of the axis length). Further, the distributions of poleward IVT for tropical and extratropical transport axes show a large degree of overlap, such that poleward IVT does not seem to offer a suitable threshold to separate these two kinds of transport axes.



Despite the mentioned differences between moisture transport axes and typical definitions of atmospheric rivers, these two
concepts generally capture the same phenomenon in the mid-latitudes. For example, the occurrence of moisture transport axes
along the North American west coast is often associated with strong precipitation (snapshot and composite analysis in Fig.
5) and the characteristic synoptic structure associated with atmospheric rivers in this region (as documented in Fig. 9 of Rutz
et al., 2019) is very similar to the mean synoptic situation conditioned on the presence of a moisture transport axes in the same
location (Fig. 5c,d).

## 6 Moisture transport axes in polar regions and their relation to warm moist intrusions

Moisture filaments and associated peaks in the moisture transport occur in a similar form also in polar regions (Woods et al.,
2013). Gorodetskaya et al. (2014) and Wille et al. (2019) discuss cases where moisture filaments made landfall on the Antarctic
coastline, referring to these features as *atmospheric rivers* to stress the similarities to their mid-latitude counterparts. Around
Antarctica, atmospheric rivers are an important factor for both liquid and frozen precipitation (Gorodetskaya et al., 2014; Wille
et al., 2019, 2021) and have also been linked to ice sheet calving events (Wille et al., 2022). When these moisture filaments
occur in the Atlantic Arctic, they would typically be called *warm moist intrusions* (e.g., Woods and Caballero, 2016; Papritz
et al., 2022), although Mattingly et al. (2018) also use the label to conceptualize moisture transport onto the Greenland Ice
Sheet. In the following, we relate detected moisture transport axes to both of these features, but focus on maritime/coastal
events, as we detect very few moisture transport axes over the ice sheets.

Returning first to the Southern Ocean and the May-2009 case of Gorodetskaya et al. (2014), a moisture transport axis
traces the essentially meridional moisture transport across all of the mid-latitudes from close to Madagascar onto the Antarctic
continent (Fig. 6a,b). Their February-2011 case is similar in that it also features a well-defined moisture transport axis across
all of the mid-latitudes, and is thus only shown in the supplement (Fig. S1c,d). In both cases, the moisture transport is well-
captured by the detected axes, but also largely by the ARTMIP schemes available for ERA5.

The synoptic structure of the May-2009 case is typical for the region (Fig. 6c,d). A composite of all cases where moisture
transport axes reach the Antarctic coastline within 200 km of the Gorodetskaya et al. (2014) case demonstrates a predominantly
meridional orientation of the axes reaching the coastline (green contours in Fig. 6d). Few transport axes penetrate into the
interior of the Antarctic continent; nearly all are diverted along the coastline (green contours and shading in Fig. 6d). In
contrast, the composite moisture transport remains largely zonal throughout most of the mid-latitudes and only exhibits a
cyclonic anomaly close to the Antarctic coast centered around 60°S (Fig. 6c). Many of these events are also captured by the
ARTMIP-ERA5 schemes (orange and pale red contours Fig. 6d), but ARTMIP detections are substantially less consistent for
this location compared to Northern California (cf. Fig. 5d; peak detection rates of 20.4% versus 28.8%).

In the Atlantic Arctic, a moisture transport axis occurs in the vicinity of Longyearbyen in between 1% and 2.5% of the winter
time steps (Fig. 3a). In line with the findings of Serreze et al. (2015), a composite analysis of these events highlights pronounced
northeastward moisture transport from the mid-latitude North Atlantic (Fig. 7). Moisture transport axes occur frequently across
the entire subpolar North Atlantic, between Greenland and Norway, with only a slight skew towards the Norwegian coast. The





composite is also fully consistent with the case study discussed in Binder et al. (2017), with pronounced meridional moisture transport between France and Fram Strait (Fig. 7a,b).

These composites and synoptic examples suggest a good correspondence between our moisture transport axes and both polar atmospheric rivers and warm moist intrusions. To corroborate this finding, we systematically compare our transport axes with the occurrence of warm moist intrusions as defined by Woods et al. (2013). They define such intrusions by a transport threshold of $200\,\mathrm{Tg\,day^{-1}\,deg.\,long^{-1}}$ across $70°\mathrm{N}$, which corresponds to about $61\,\mathrm{kg\,s^{-1}\,m^{-1}}$. We relate this to the poleward IVT along all moisture transport axes detected between 68 and $72°$ latitude (Fig. 8c). We include detections within $\pm2°$ latitude to increase the sample size for this analysis.

About 60% of the transport axes exceed this threshold (Fig. 8c). Further, a skew in the poleward IVT distribution towards positive values shows that moisture transport is predominantly poleward. Nevertheless, about 20% of moisture transport axes around $70°$ latitude exceed a threshold of the opposite sign, signaling the regular occurrence of warm moist *extrusions* from the Arctic. This phenomenon has also been noticed by Papritz et al. (2022), who separated total and net moisture transport into the Arctic. Such emphasized moisture export from the Arctic is plausible from the synoptic example in Fig. 2f. In this snapshot, a moisture transport axis turns equatorward around $60°\mathrm{S}$, tracing the warm front of a mature cyclone. An analogous synoptic situation with a cyclone core located in the Barents Sea would simultaneously yield strong moisture import into and export from the Arctic.

Considering transport irrespective of direction, few transport axes feature less than $200\,\mathrm{kg\,s^{-1}\,m^{-1}}$ (Fig. 8b). Further, while most polar transport axes feature TCWV below $20\,\mathrm{kg\,m^{-2}}$, it is interesting to note the occasional occurrence of moisture transport axes with up to around $40\,\mathrm{kg\,m^{-2}}$ even at around $70°$ latitude (Fig. 8a).

## 7 Moisture transport axes in the subtropics and tropics

Potential extensions of the concept of atmospheric rivers towards the subtropics have not yet been considered systematically. Given that the Hadley circulation exhibits much less day-to-day variability than the extratropics, such that subtropical climate is generally well-described by monthly mean states (e.g., Vallis, 2006; Ogawa and Spengler, 2019), this disregard might be well-justified. In contrast, atmospheric rivers are considered a synoptic feature with a life cycle on the time scale of days, that is usually thought to occur in conjunction with an extratropical cyclone (Dacre et al., 2015; Azad and Sorteberg, 2017). It is thus a-priori unclear to what extent the atmospheric river concept remains physically meaningful at lower latitudes. Nevertheless, well-defined maxima in the moisture transport, i.e. moisture transport axes, do exist in the subtropics and tropics (cf., snapshot in Fig. 2g,h; climatologies in sec. 4 and Fig. 3).

The low-latitude case study and climatologies discussed previously suggest different dynamical reasons for the occurrence of moisture transport axes in this region. Some seem to be steered by orography (e.g., across the Maritime Continent and along the Andes), some seem to capture actual intermittent moisture transport similar to that in the extratropics (e.g., Sahel region) and some seem to capture seasonal circulation anomalies like the Indian Monsoon. The latter result is in line with Park et al. (2021), who use atmospheric rivers to conceptualize variability in the East Asian Summer Monsoon. Interestingly, the



topographically steered moisture transport axes mostly vanish with normalization (cf. Fig. S2 in the supplement), implying an almost continuous moisture transport in these regions. In contrast, the Indian and East Asian monsoons remain visible with normalization by the annual means (Fig. S2), but do largely vanish when normalizing by the seasonal mean (not shown) due to the near-stationary flow during the monsoon season.

Using composite analyses, we contrast the moisture transport axes detected in the near-stationary Indian Monsoon with the
intermittent transport axes detected over the Sahel region (Fig. 9). The composite analysis for the Indian Monsoon is based on the occurrence of transport axes in the vicinity of Kolkata, India. This composite comprises 8982 time steps, corresponding to about 56 days per year, highlighting the semi-permanent nature of the feature during the monsoon season (Fig. 9b). The composite moisture transport axes are diverted by the Himalayas, with westward-pointing moisture transport axes along the mountain range to the west of the Bay of Bengal, and eastward axes to the east (Fig. 9b). This diffluence is not visible in
the direction of the mean transport (Fig. 9a), because the magnitude of the mean transport vector is relatively small along the Himalayas (gray contours in Fig. 9b). In combination with the high TCWV evident in Fig. 9a, this indicates a variable transport direction along the mountain range.

With our second composite analysis, we move from one of the (seasonally) most humid to one of the most arid places in the subtropics (Fig. 9c,d). Here, the composites are based on the occurrence of moisture transport axes close to Timbuktu, Mali,
which is located in the Sahel belt to the south of the Sahara Desert. The snapshot in Fig. 2g,h includes an example case included in this composite. In line with the snapshot, the composite structure shows a tilted extrusion of moisture from the ITCZ into the Sahel region (Fig. 9c). Although the average precipitation along these transport axes is not strong enough to exceed the contour level, the snapshot in Fig. 2g,h illustrates that scattered precipitation can be associated with such transport events.

In the Sahel region, the occurrence of moisture transport axes is very intermittent, mainly occurring during late summer
(JAS) and winter to early spring (DJFMA). The composite includes about 16 days per year. Their occurrence during JAS coincides with the seasonal occurrence of African Easterly Waves (e.g., Berry et al., 2007). In contrast, the occurrence during winter and spring might be related to an influence from the mid-latitudes due to the southward displacement of the stormtrack (e.g., Spensberger and Spengler, 2020).

## 8 Summary and concluding remarks

We introduced a novel and generic feature detection algorithm for moisture filaments using an algorithm developed to detect upper tropospheric jets (Spensberger et al., 2017). We call the detected features *moisture transport axes* as these trace lines of maximum vertically integrated moisture transport (IVT). The name *moisture transport axes* and the visualization as lines are not meant to imply a continuous transport along the line. As Läderach and Sodemann (2016) and Dacre et al. (2019) pointed out, there is a continuous recycling of water along atmospheric rivers, even though some long-range transport of moisture does
occur (Stohl et al., 2008; Sodemann and Stohl, 2013). Analogously, we expect the same to be true for moisture transport along our detected moisture transport axes.



In the mid-latitudes, moisture transport axes generally capture the same synoptic phenomenon as commonly used definitions of atmospheric rivers. Due to the structure-based definition of the feature, moisture transport axes often trace the moisture filament further into the subtropics, continents, or polar regions than the ARTMIP detections available for ERA5. Further,

moisture transport axes reveal the substructure of atmospheric rivers with several distinct maxima in the IVT and they are not subject to any limitations in the orientation of the moisture transport.

For polar regions (except the ice sheets), we document a relation to warm moist intrusions and to existing polar adaptations of atmospheric rivers. Moisture transport axes thus highlight events with pronounced moisture import into polar regions. They, however, also reveal synoptic structures where much of the imported moisture is directly exported again. While such events

are implicitly accounted for in the analysis of Papritz et al. (2022), the existence of such moisture export events is not obvious from previous studies on warm moist intrusions.

In the tropics and subtropics, moisture transport axes highlight both intermittent, seasonal and near-stationary features of the circulation. For example, moisture transport axes both highlight the continuously strong moisture transport in the Indian Monsoon circulation or the more transient peaks in the monsoon circulation. In the climatologically arid Sahel region, moisture

transport axes reflect intermittent extrusions of moisture from the deep tropics. Finally, moisture transport axes pick up the orographically steered South American low level jet as well as moisture transport along the straits of the Maritime continent. While these spotlights are far from providing a comprehensive view of subtropical moisture transports, they clearly suggest that the concept of moisture transport axes remains meteorologically meaningful also in these regions.

In conclusion, our approach allows us to unify atmospheric rivers, warm moist intrusions, and monsoon air streams into one

common concept: *moisture transport axes*. As our definition is based on the typically elongated structure of moisture transports, our detection algorithm performs seamlessly from the tropics across the mid-latitudes into the polar regions. The concept of transport axes might thus turn out to be particularly useful to study moist interactions between the tropics and subtropics, mid-latitudes, and polar regions. Further, our definition of moisture transport axes is likely more robust across varying climates than commonly used definitions of atmospheric rivers, as our definition might not require changes to thresholds or a time-

dependent normalization. Finally, following the approach of Spensberger and Spengler (2020), moisture transport axes enable the investigation of variability in the occurrence of atmospheric rivers largely independent from their varying intensity.

*Code and data availability.* The ERA5 reanalysis (Hersbach et al., 2023a, b) and corresponding ARTMIP detections (Atmospheric River Tracking Method Intercomparison Project, 2022) used in this study are all publicly available. The dataset of standard and normalized MTAs 1979-2020 will be published with the acceptance of the manuscript for publication. The jet detection algorithm is available as part of *dynlib*,

a library of meteorological analysis tools (Spensberger, 2024).

*Author contributions.* CS conducted the analyses, visualised the results, and did the initial writing. All authors contributed to the conceptual development of the method and refining the writing.



*Competing interests.* We declare no competing interests.

*Acknowledgements.* TWe thank ECMWF for providing the ERA5 reanalysis used in this study. The reanalysis was obtained directly through

the Meteorological Archival and Retrieval System (MARS).



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



**Figure 2.** Snapshots of moisture transport axis occurrence. Panels (a-d) show atmospheric river cases discussed by Lora et al. (2020) by 5 November 2006, 09Z, and 14 November 2006, 09Z, respectively, (e,f) the strongest river case listed in Azad and Sorteberg (2017) on 14 September 2005, 00Z, and (g,h) a moisture transport axis over the Saharan Desert on 15 December 2017, 06 UTC. The left column shows total column water vapor [kg/m$^2$] (shading), IVT (arrows), and total precipitation (pink contour, 1 mm/h). The right column shows the magnitude of IVT [kg m/s]. Black contours show sea-level pressure below 1010 hPa in steps of 5 hPa. Red, orange and pale red contours show the number of atmospheric river detections in the ARTMIP-ERA5 dataset with contours at 1, 3, and 6 of 6 available global algorithms. Finally, the yellow lines show detected moisture transport axes.



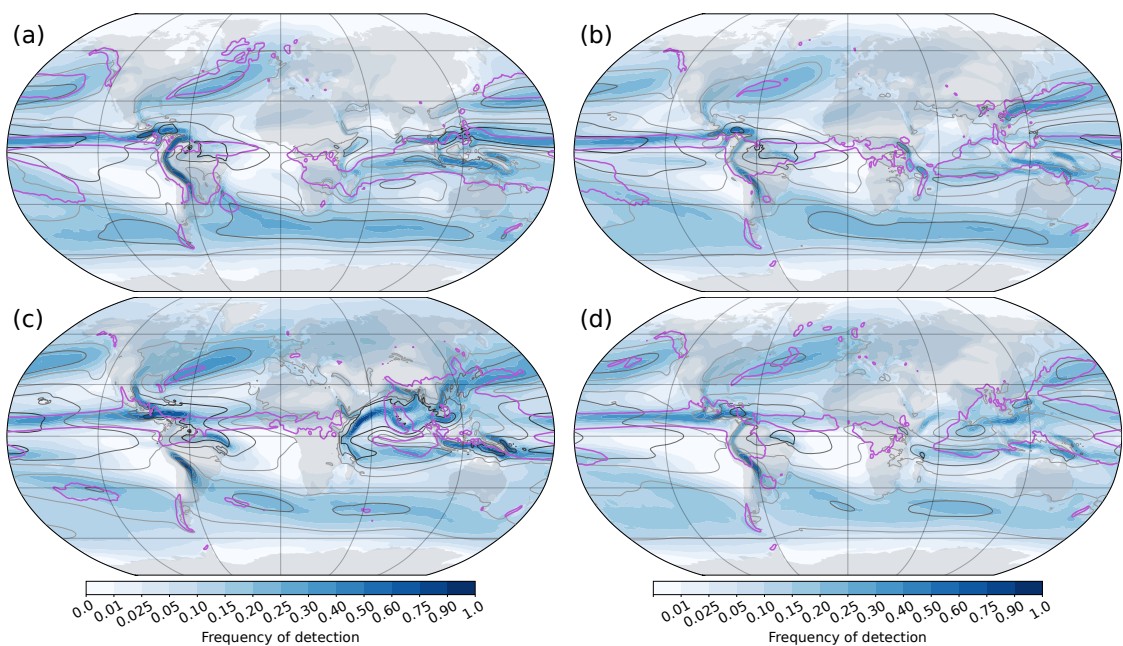

**Figure 3.** Climatological occurrence of moisture transport axes within 200 km of any given location for (a) DJF, (b) MAM, (c) JJA, and (d) SON. Grey contours show the climatological magnitude of the vertically integrated water vapor transport with contours at 100, 200, and 400 kg/(m s) and the pink contour marks the region exceeding 5 mm/day of total precipitation.



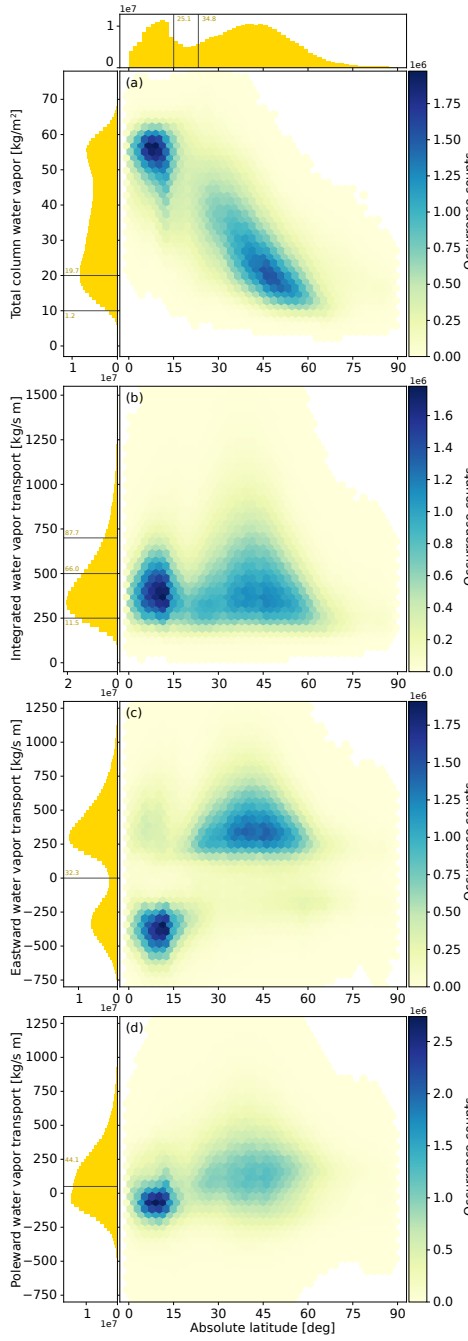

**Figure 4.** Two-dimensional histograms showing the occurrence counts of moisture transport axes in phase spaces defined by latitude and (a) total column water vapor and (b) magnitude of IVT, as well as (c) eastward and (d) poleward component of IVT. One-dimensional histograms along the respective axes are displayed in yellow along the sides. Typical detection thresholds used for atmospheric river detections are indicated by lines across the one-dimensional histograms, with the adjacent numbers indicating the percentile at which these lines occur in the respective distributions.

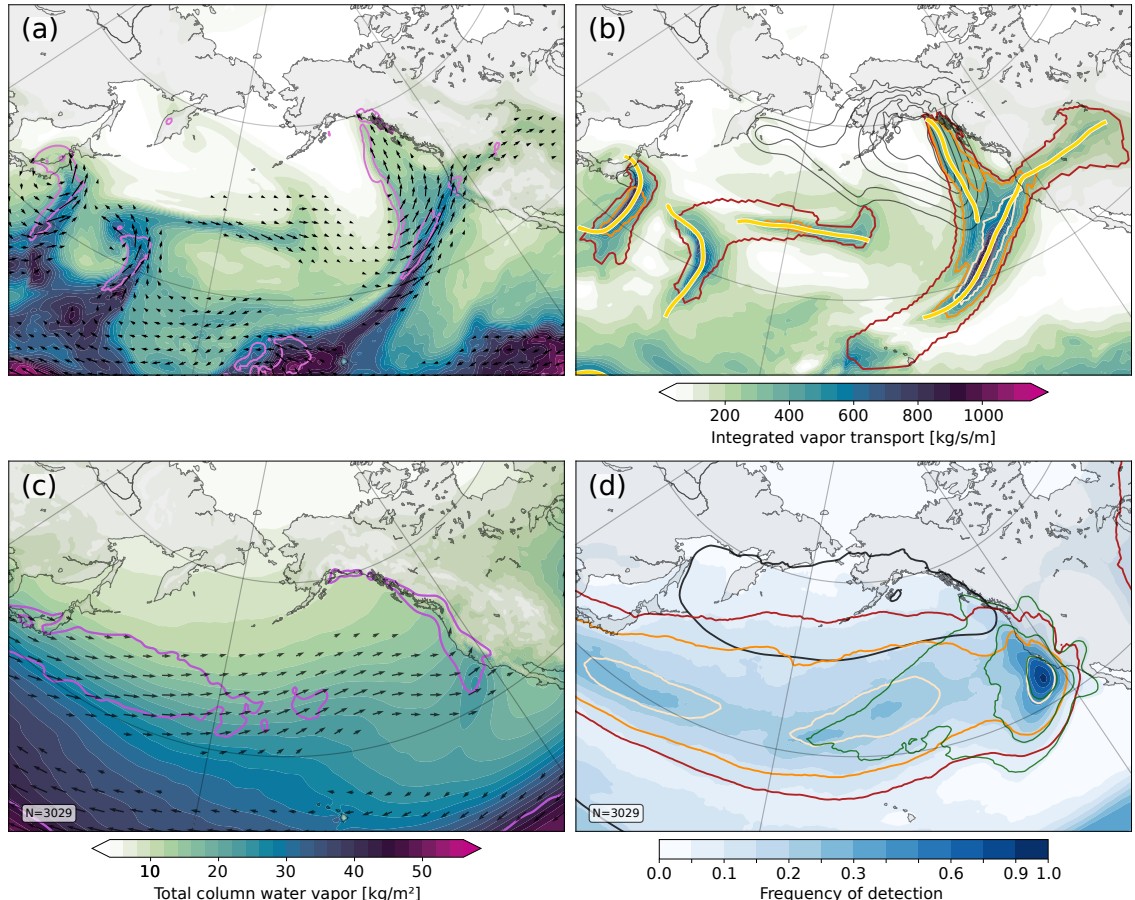

**Figure 5.** (a,b) Snapshot of an atmospheric river case from 15 February 2014, 09Z, discussed in Rutz et al. (2019) analogous to each row in Fig. 2. (c,d) Composites based on the occurrence of a moisture transport axis within 200 km of the position 39°N, 124°W (cf. Fig. 9 of Rutz et al., 2019). (c) Total column water vapor [kg/m$^2$] (shading) with the gray contour highlighting the 20 kg/m$^2$ contour, and IVT (arrows), and total precipitation (pink contour at 5 mm/day). (d) Frequency of occurrence of normalized moisture transport axes within 200 km. Dark green contours show the frequency of occurrence for only those moisture transport axes that intersect the target region (contours at 0.05, 0.1, 0.2, 0.4, 0.6 and 0.9). Black contours show sea-level pressure below 1010 hPa in steps of 5 hPa. Finally, red, orange and pale red contours show the frequency of atmospheric river detections for the 6 schemes available for ERA5 with contours at 5%, 10%, and 20%, respectively. The composites are based on the period 2000-2019 as ARTMIP-ERA5 detections are generally only available for this period. For all other variables composites for 1979-2020 would be nearly indistinguishable (not shown).

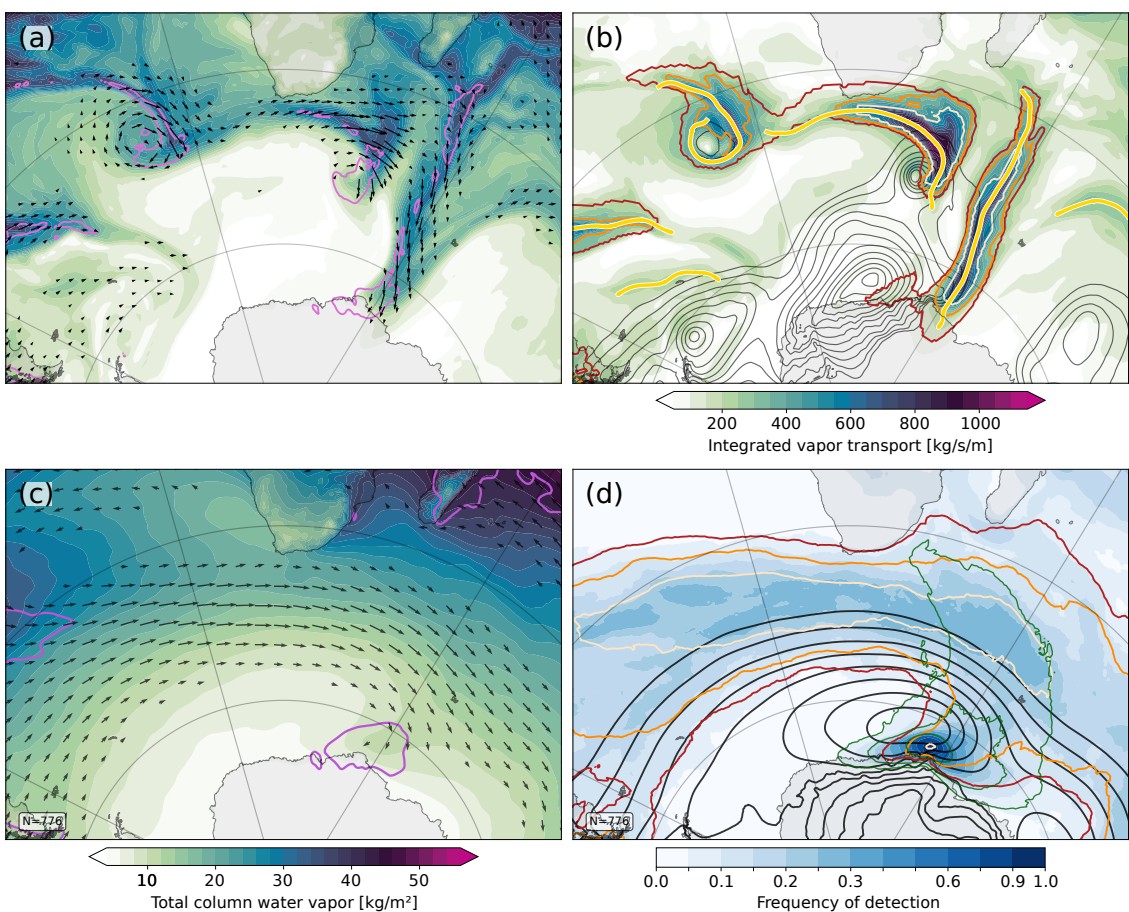

**Figure 6.** As Fig. 5, but for atmospheric rivers making landfall on Antarctica. The snapshot from 19 May 2009, 00Z in (a,b) is one of the cases discussed in Gorodetskaya et al. (2014). The composite (c,d) is based on moisture transport axes occurring within 200 km of the landfall location.



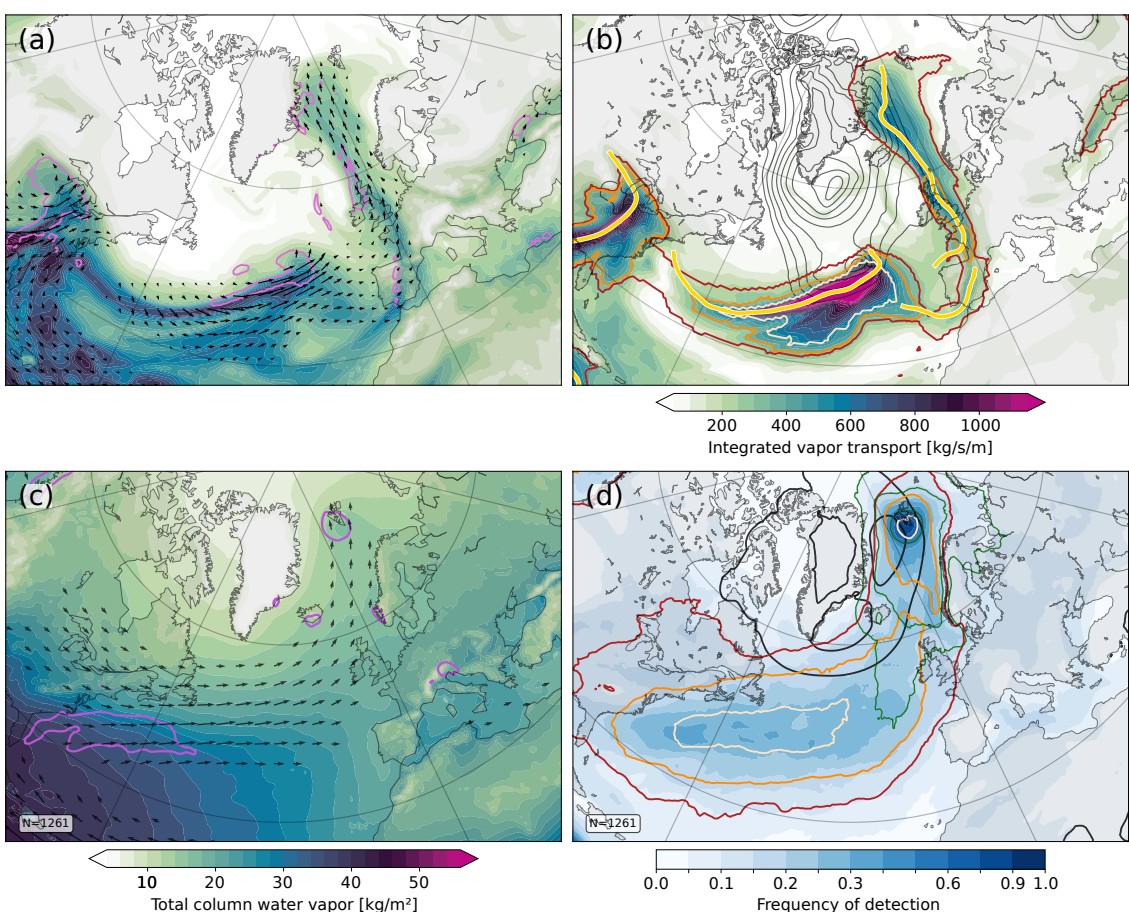

**Figure 7.** As Fig. 5, but for moisture transport axes around Svalbard. The snapshot from 29 December 2015, 00Z, in (a,b) is discussed in Binder et al. (2017). The composite (c,d) is based on moisture transport axes occurring within 200 km of Longyearbyen, Svalbard.



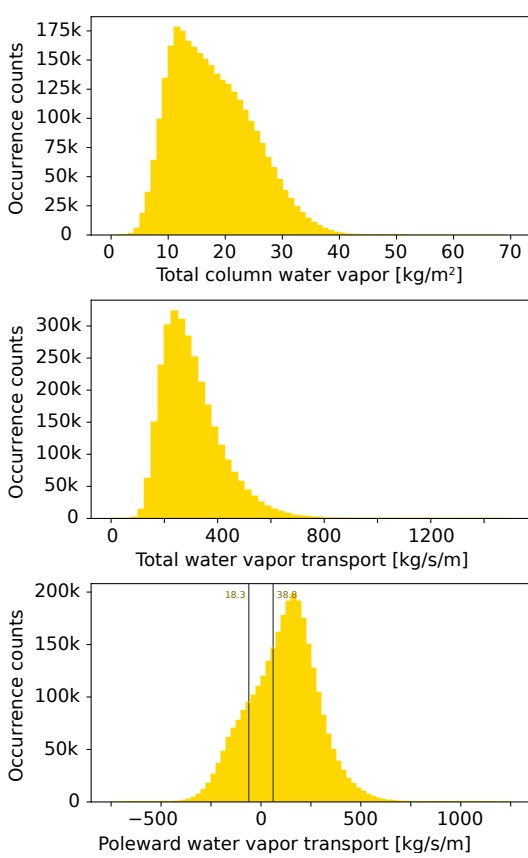

**Figure 8.** As the one-dimensional histograms on the left-hand side of Fig. 4a,b and d, but based on moisture transport axes detected within 68-72°N/S. As in Fig. 4, yellow and red histograms represent non-normalized and normalized detections, respectively. For the relevance of the vertical lines and percentiles in (c) refer to the main text.





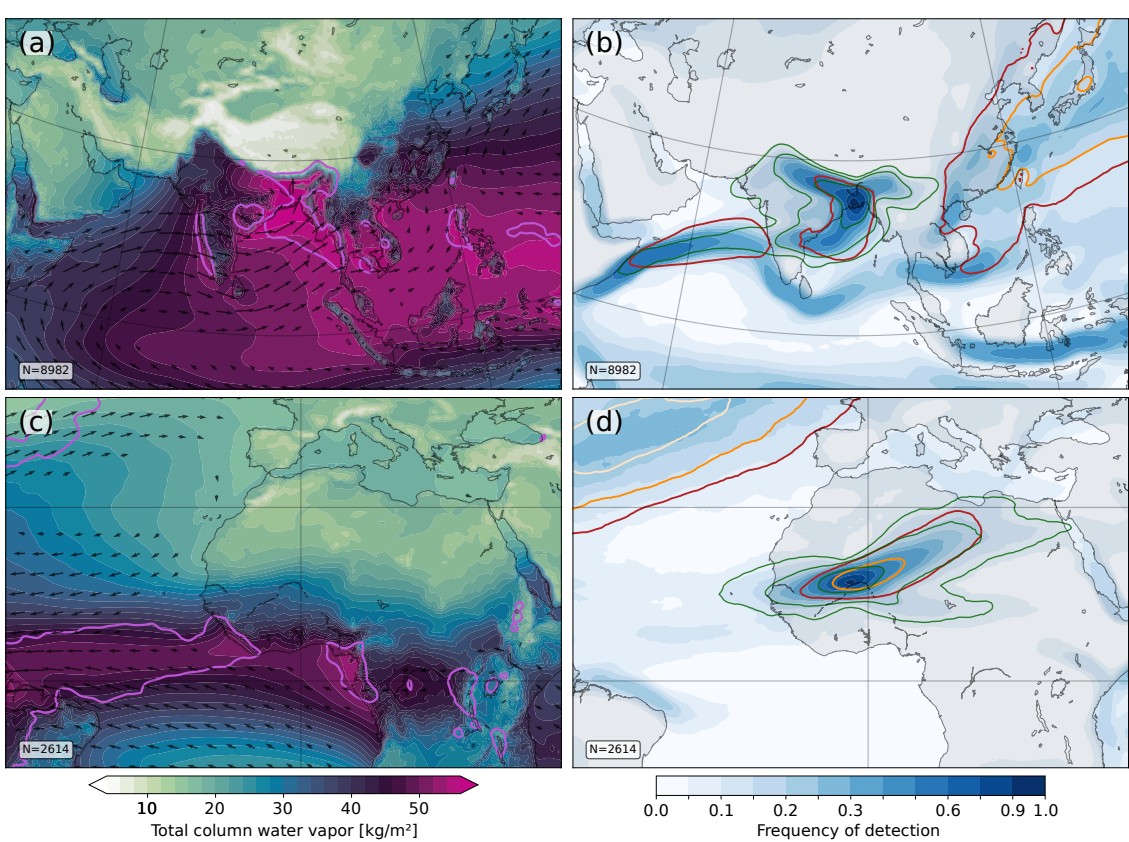

**Figure 9.** As the composites in Fig. 5c,d, but (a,b) for moisture transport axes within 200 km of Kolkata, India, and (c,d) for moisture transport axes within 200 km of Timbuktu, Mali, in the Sahel region