# Peer review of "Moisture transport axes: a unifying definition for tropical moisture exports, atmospheric rivers, and warm moist intrusions"

_EGUsphere, 2024_

## Referee Comment (RC1)

**Review of "Moisture transport axes: a unifying definition for monsoon air streams, atmospheric rivers, and warm moist intrusions" by Spensberger et al. submitted to WCD**

Thank you for giving me the opportunity to read and review this article on the detection of moisture transport axes. I found it well-written and beautifully illustrated. I really like the proposed new Eulerian method based on one-dimensional moisture transport axes, i.e. lines connecting local maxima in moisture transport. This latitude-independent method allows to identify intense horizontal moisture transport in the atmosphere in an elegant and computationally efficient way, with several advantages compared to other approaches. The advantages I see are: 1) concept-unifying character, 2) region-independent definition, 3) its flexibility with respect to orientation (e.g. no requirement of a westward orientation from a given source region).

I have one major concern with the writing, which is related to the disproportionate focus on atmospheric rivers in the introduction and the slight (certainly unwanted) neglect of the existing tropical and subtropical literature:

1) I found the abstract very convincing in the way the new method is introduced and highlighted as concept-unifying. Really great abstract! But in the introduction, I desperately missed this elegant way to approach the topic of detecting "narrow elongated filaments" comprehending atmospheric rivers, tropical moisture exports, monsoon air stream, warm conveyor belts, warm moist intrusions into polar regions. The introduction focusses entirely on atmospheric rivers. That's a pity, because this fact touches on the characterising advantage and the selling argument for this new method. To introduce moisture transport axes as an atmospheric river detection scheme, raises the question: what is the advantage of this method compared to others? And you bravely address this point at the end of the introduction. There are some practical advantages, indeed. But is your method really just yet another atmospheric river detection scheme? There are so many, and we can reasonably detect these features with the existing methods. In my view, your method is more than that. It's really cool that you can approach the detection of intense moisture transport features from such a "simple" geometric point of view, which has the potential to bring together many different concepts that were either formulated in a region-specific, impact-specific way, or which involves more sophisticated (and computationally expensive) Lagrangian detection methods.

So in short, I would recommend a rewriting of the introduction in the following way:

- There are different concepts in the meteorological literature that all characterise narrow elongated filaments of enhanced moisture transport (or total column moisture content), when going from the tropics to the poles: tropical moisture exports (transient features), monsoon air streams (more persistent), atmospheric rivers, cyclone feeder airstreams, warm conveyor belts, warm moist intrusions into polar regions. Some of these concepts overlap, others are distinct due to their regional extent, or due to the specificities of their detection, see *Ralph, Dettinger, Rutz and Waliser, Atmospheric Rivers: Chapter 2, Sodemann et al. 2020: Structure, process, mechanisms,* https://link.springer.com/book/10.1007/978-3-030-28906-5.

- Give a short overview of groups of detection schemes, staying rather general to keep the readership large. What do the existing groups of detection schemes target, resp. what do they miss by definition. Maybe a distinction between Eulerian and Lagrangian methods would also help to put forward the advantages and limitations of the new method? Why is the literature about Lagrangian airstream detection avoided and completely omitted (except for the short mention of warm conveyor belts in the abstract)? To me the moisture transport axes have several useful and complementary advantages compared to Lagrangian air stream definitions.
- Shift the method-specific and rather detailed part of the introduction at L. 46-71 to the method section
- Clearly define the goal of the study: why is it important to detect moisture filaments and why is such a concept-unifying definition useful?
- For the tropical literature I think Knippertz (2007) gives a nice synthesising overview of phenomena at low latitudes.
- In relation to high impact weather the climatology of de Vries (2022) might be relevant. Something that his not mentioned at all but that I find relevant is that the moisture transport axes method focusses on the climatology of coherent, large-scale enhanced moisture transport in the atmosphere but not restricted to precipitation extremes. This allows to study climatological features of moisture transport in the atmosphere and their changes with global warming.

In addition, I have the following minor comments:

2) L.1: could you omit "**extratropical"** and write "**horizontal** water vapour transport is mainly organised in narrow elongated filaments"? Isn't your method globally applicable, as long as the moisture transport is coherent and of a certain minimum spatial extent? I just ask, because I believe your detection method also includes features like tropical moisture exports (Knippertz and Wernli, 2010) and monsoon features, which I would not classify as extratropical.

3) L. 3: I know the "feeder air stream" is the official name in the Dacre et al. studies but for a more general readership, I would find it useful to call it "**midlatitude cyclone** feeder air stream".

4) L. 10: this is really nice because it illustrates the convergence needed to form a strong atmospheric river and further develops the analogy to a river system as defined by the confluence of streams in a catchment. In this sense the moisture transport axis method allows to characterise the substructure of an atmospheric river in a physically meaningful way.

5) I think the paper would benefit from an additional sentence at the end of the abstract to point out what the scientific learnings and broader implications from this study are.

6) L. 24: do the definitions diverge widely or differ in the details?

7) L. 67-71: It reads a bit strange to have "First," and then "Finally". "First," calls for "Second,".

8) L. 74: It requires some effort for me to convert T84 into a length scale in degree or km, if possible, make it easier for readers like me.

9) L. 90: I don't really understand why you do need to include "IVT absolute magnitude" in your threshold. Initially, while reading up to here, I enthusiastically thought that your method would avoid exactly that. Now why do you introduce |IVT| when filtering

out "weak" maxima, nevertheless? Certainly, this is going to remove many moisture transport axes in polar regions. What happens if you omit this and just filter out minima? You get lots of spurious axes? And you cannot filter them out with the minimum length? Do you then get spaghetti-like messy axes? I am very curious about this and would like to understand this better.

10) L. 95: It was exactly when thinking about long climate model simulations that I thought that a |IVT|-independent threshold (i.e. one that would just filter out minima) would be very valuable.

11) L. 107: Is this similar to what is done in front detection? Are there any parallels to front detection schemes in your method that would be worthwhile mentioning?

12) L. 119: When looking at your case studies, I note that the features you detect with the moisture transport axes are coherent large-scale phenomena which are of meteorological relevance. Some properties of the detected moisture transport axes also reveal meso-scale features in addition, which are to date not well studied but which are likely relevant for the understanding and adequate prediction of these systems (among others atmospheric rivers) and their impact.

13) L. 127: does this relate to the two types of WCB outflow branches (cyclonic and anticyclonic outflow?) see Heitmann et al. 2024.

14) L. 132 and elsewhere: "pick up" sounds a bit like slang. Can you reformulate?

15) L. 133: "picked up by one to three of the six global detection algorithms" I don't understand what this means. Why writing "one to three" and not mentioning a clearly defined number of algorithms?

16) L. 137: "expulsion of Tropical moisture" and at L. 320: "extrusions of moisture from the deep tropics". I would use an existing term such as tropical moisture export.

17) L. 131-140: Here you already compare your method to existing atmospheric river detection schemes, and you do this sort of evaluation again in Section 5. I would recommend to group all the comparison effort to existing methods in one method evaluation section, which I would personally prefer to have before the more phenomenological and scientific discussion about the relationship of moisture transport axes to different tropical, midlatitude and polar features.

18) L. 176: MTA is not defined, and I preferred the written out version "moisture transport axes".

19) L. 202: off -> of

20) L. 206-207: I don't understand this sentence.

21) Section 5: I find the discussion around the detection of atmospheric rivers very technical and sometimes difficult to follow. The panel d in Figures 5-9 only show a few contours from the existing atmospheric river detection in the climatological plots and the chosen way of illustrating this intercomparison makes it difficult to compare the new method quantitatively with existing ones. If possible, I would separate the evaluation of the method based on a comparison to others from the scientifically interesting discussion of what the new method detects and what we can learn from it about atmospheric river dynamics.

22) L. 262-269: I find this way of approaching the identified tropical phenomena a bit awkward. Indeed, there are more persistent features linked to the Monsoon systems but there are also many transient features such as tropical plumes (Rubin et al. 2007) or tropical moisture exports (Knippertz, 2007), often times these systems are related to Rossby wave breaking and are relevant for extreme precipitation in the subtropics (de Vries, 2021).

23) L. 286-287: are these barrier winds? What does the direction of transport depend on? Are these relevant questions for forecasting in these regions? And can you see and propose how one could address them by using moisture transport axes?

24) L. 295: I think here one could establish a link with the West African Monsoon (Fink et al. 2017).

25) L. 297: Yes, indeed, that is also, when most precipitation falls in the Sahara (Armon et al. 2024).

26) Fig. 3 and others: Probably it is just my printer but the delimitation of the continents is barely (not) visible and makes it a bit difficult to orient

27) L. 305 recycling vs. large-scale transport: A large share of moisture in cyclone precipitation is fed through the cyclone feeder airstream and originates from the cold sector of previous cyclones as well as the cyclone-anticyclone interaction zone (see, Papritz et al. 2021). Could moisture transport axes be combined with cyclone masks and tracks to study the cyclone-to-cyclone moisture hand over and multi-cyclone association of intense moisture transport in more detail? What about the temporal evolution of moisture transport axes? Can two subsequent moisture transport axes be related to each other?

28) L. 307-311: Yes, the tracing of the moisture filaments further into the subtropics, continents and/or polar regions is a very nice characteristic of the new method but here I think the Eulerian vs. Lagrangian aspect should be mentioned and discussed as a caveat resp. as a possible outlook: this of course does not mean that you suggest that moisture transport in the atmosphere is generally occurring over longer distances or timescales. A combination with a trajectory-based diagnostic would be required for investigating the moisture cycling aspect along these moisture filaments.

29) L. 311: yes, and I think this is really exciting because it allows to study the substructure of atmospheric rivers in more detail, in particular the relevance of moisture recycling through precipitation evaporation and the importance of cold pool-like circulations within the complex cloud-systems.

30) L. 320: "extrusions of moisture from the deep tropics"->tropical moisture exports? Or is the feature you describe something meteorologically different?

31) L. 324-331: yes, exactly, very nice and convincing concluding paragraph, that's the framing I would also strongly encourage to adopt for the introduction.

32) I miss a serious discussion of the caveats of the method in the conclusions and an outlook about which scientific questions could be addressed with this new valuable detection scheme.

33) L. 339: TWe -> We

34) L. 333: I commend the authors on their plan to make their climatology of moisture transport axes publicly available. MTAs have not been introduced as an abbreviation -> write it out?

**References:**

Armon, M., de Vries, A. J., Marra, F., Peleg, N., and Wernli, H. (2024): Saharan rainfall climatology and its relationship with surface cyclones, Weather and Climate Extremes, 131, 100 638, https://doi.org/10.1016/j.wace.2023.100638.

de Vries, A. J. (2021): A global climatological perspective on the importance of Rossby wave breaking and intense moisture transport for extreme precipitation events, Weather Clim. Dynam., 2, 129–161, https://doi.org/10.5194/wcd-2-129-2021.

Fink, A. H., Engel, T., Ermert, V., Van Der Linden, R., Schneidewind, M., Redl, R., et al. (2017). Mean climate and seasonal cycle. In Meteorology of tropical West Africa: The forecasters' handbook (pp. 1–39). Chichester, UK: John Wiley & Sons, Ltd. https://doi.org/10.1002/9781118391297.ch1

Heitmann, K., Sprenger, M., Binder, H., Wernli, H., and Joos, H. (2024): Warm conveyor belt characteristics and impacts along the life cycle of extratropical cyclones: case studies and climatological analysis based on ERA5, Weather Clim. Dynam., 5, 537–557, https://doi.org/10.5194/wcd-5-537-2024.

Knippertz P (2007): Tropical–extratropical interactions related to upper-level troughs at low latitudes. Dyn Atmos Oceans 43:36–62. https://doi.org/10.1016/j.dynatmoce.2006.06.003

Knippertz, P., and H. Wernli (2010): A Lagrangian Climatology of Tropical Moisture Exports to the Northern Hemispheric Extratropics. J. Climate, 23, 987–1003, https://doi.org/10.1175/2009JCLI3333.1.

Papritz, L., F. Aemisegger, and H. Wernli (2021): Sources and Transport Pathways of Precipitating Waters in Cold-Season Deep North Atlantic Cyclones. J. Atmos. Sci., 78, 3349–3368, https://doi.org/10.1175/JAS-D-21-0105.1.

Ralph, Dettinger, Rutz and Waliser (2020): Atmospheric Rivers, https://link.springer.com/book/10.1007/978-3-030-28906-5#about-this-book. And from this book I particularly recommend:

Sodemann, Wernli, Knippertz, Corderia et al., Chapter 2: Structure, Process, and Mechanism.

Rubin, S., B. Ziv, and N. Paldor (2007): Tropical Plumes over Eastern North Africa as a Source of Rain in the Middle East. Mon. Wea. Rev., 135, 4135–4148, https://doi.org/10.1175/2007MWR1919.1.

---

## Author Response (AR1)

**Response to reviewers – "Moisture transport axes: a unifying definition for monsoon air streams, atmospheric rivers, and warm moist intrusions"**

C. Spensberger, K. Konstali & T. Spengler

7 October 2024

We sincerely thank Franziska Aemisegger and an anonymous reviewer for the constructive and in-depth review of our work. We are very happy to read that both reviewers found the material to be well-presented, and even more so that we managed to convey what we believe are the novel aspects of our feature detection approach. Our point-by-point response to the issues raised appears below in blue.

**Reviewer 1 (F. Aemisegger)**

**1)** I have one major concern with the writing, which is related to the disproportionate focus on atmospheric rivers in the introduction and the slight (certainly unwanted) neglect of the existing tropical and subtropical literature. [...]

We sincerely thank you for this comment. We are very happy to read that the reviewer regards our approach to detecting moisture filaments to be more than yet another atmospheric river detection scheme. This is very much the way in which we see the work ourselves.

The work was previously submitted at JGR-A, where the five anonymous reviewers all seemed to belong to the atmospheric river community. To address their comments, we were forced to put much more emphasis on the relation of our detection algorithm to atmospheric rivers than we originally intended. We are thus very happy to shift focus away from atmospheric rivers back to the novel aspects of our detection approach. We reworked the introduction following the reviewer's recommendations.

We also thank for the additional context on tropical moisture transport, which we happily included in the discussion in this section. We further removed the introductory paragraph to this section, and found the remainder to work well without any replacement.

**Minor comments**

**2)** L. 1: could you omit "extratropical" and write "horizontal water vapour transport is mainly organised in narrow elongated filaments"? Isn't your method globally applicable, as long as the moisture transport is coherent and of a certain minimum spatial extent? I just ask, because I believe your detection method also includes features like tropical moisture exports (Knippertz and Wernli, 2010) and monsoon features, which I would not classify as extratropical.

We agree, removed "extratropical" and rephrased following your suggestion.

**3)** L. 3: I know the "feeder air stream" is the official name in the Dacre et al. studies but for a more general readership, I would find it useful to call it "midlatitude cyclone feeder air stream".

We see the reviewer's point, but decided to not implement this change. If adding this specification, we should for consistency also add it for WCBs, and that would make the list a bit cumbersome to read. But as a side remark, we did add "tropical moisture exports" to this list of related names and concepts, reflecting the suggested changes to the introduction.

**4)** L. 10: this is really nice because it illustrates the convergence needed to form a strong atmospheric river and further develops the analogy to a river system as defined by the confluence of streams in a catchment. In this sense the moisture transport axis method allows to characterise the substructure of an atmospheric river in a physically meaningful way.

We are happy the reviewer saw this point and agrees with us on its significance!

**5)** I think the paper would benefit from an additional sentence at the end of the abstract to point out what the scientific learnings and broader implications from this study are.

Thanks for the suggestion. We agree and added the sentence "As our detection algorithm performs seamlessly from the tropics across the mid-latitudes into the polar regions, our approach might turn out to be particularly useful to study moist interactions between the tropics and subtropics, mid-latitudes, and polar regions." to the end of the abstract.

**6)** L. 24: do the definitions diverge widely or differ in the details?

We would argue the definitions diverge widely, as there is no consensus on (a) whether to base the detection on moisture content or transport, (b) whether to use absolute or relative thresholds, and (c) on whether to impose geometrical constraints. These questions are more than details.

**7)** L. 67-71: It reads a bit strange to have "First," and then "Finally". "First," calls for "Second,".

We agree and removed the "First," and replaced the "Finally," with "In addition,".

**8)** L. 74: It requires some effort for me to convert T84 into a length scale in degree or km, if possible, make it easier for readers like me.

We agree and now mention the equivalent resolution of approximately 150 km.

**9)** L. 90: I don't really understand why you do need to include "IVT absolute magnitude" in your threshold. Initially, while reading up to here, I enthusiastically thought that your method would avoid exactly that. Now why do you introduce —IVT— when filtering out "weak" maxima, nevertheless? Certainly, this is going to remove many moisture transport axes in polar regions. What happens if you omit this and just filter out minima? You get lots of spurious axes? And you cannot filter them out with the minimum length? Do you then get spaghetti-like messy axes? I am very curious about this and would like to understand this better.

Interesting idea. Our approach was inspired from the jet detection algorithm in Spensberger et al. (2017), where we require the wind maximum to be "well-defined". Sharp wind maxima at low wind speeds would generally not be regarded as jets, so it was natural to have absolute wind speed contribute to the threshold.
  Analogous to that reasoning, we do not impose an |IVT|-threshold separately, but only a combination of |IVT| and the shear gradient $\frac{\partial \sigma_{IVT}}{\partial n}$. Removing the |IVT|-contribution from this field would imply: (a) IVT-maxima of intermediate sharpness will be detected also along very weak moisture filaments, (b) somewhat diffuse maxima along very strong moisture filaments might no longer be detected. In addition to detecting a number of moisture features that are irrelevant for weather we might thus also no longer detect some unambiguous atmospheric river cases, for example when a secondary cyclogenesis along a cold front locally distorts the structure of the (otherwise strong) moisture transport.

**10)** L. 95: It was exactly when thinking about long climate model simulations that I thought that a |IVT|-independent threshold (i.e. one that would just filter out minima) would be very valuable.

We have applied our algorithm to climate model simulations and found the algorithm to work well without adaptations for simulations with CESM2 covering the period 1960-2100 (Konstali et al., 2024a). There was unfortunately no preprint available for this work, but it has recently been published. We thus included a reference to Konstali et al. (2024a) in the final publication of this manuscript, supporting our claim.

**11)** L. 107: Is this similar to what is done in front detection? Are there any parallels to front detection schemes in your method that would be worthwhile mentioning?

True, front line detections in principle need to solve the same normalisation issue for their climatologies. However it seems like published front climatologies generally ignore the issue, and only show the (un-normalised) frequency of lines within each grid point (e.g. Jenkner et al., 2010; Berry et al., 2011; Schemm et al., 2015).
  The frequencies in these climatologies thus depend on the (local) grid cell size, with double the cell size yielding double the front frequency. This spuriously reduces detection frequencies at high latitudes on a lat-lon grid, and makes it hard to compare climatologies across datasets of varying resolution.
  We were not aware of the problem yet in Spensberger et al. (2017) and there also show un-normalised jet detection frequencies. We later solved the issue in Spensberger and Spengler (2020), but the resulting

unit of line length per unit area turned out to be causing confusion. Since then we have been using the approach outlined here.

**12)** L. 119: When looking at your case studies, I note that the features you detect with the moisture transport axes are coherent large-scale phenomena which are of meteorological relevance. Some properties of the detected moisture transport axes also reveal meso-scale features in addition, which are to date not well studied but which are likely relevant for the understanding and adequate prediction of these systems (among others atmospheric rivers) and their impact.

Great, these are exactly the points we would like readers to note in these snapshots! The focus on coherent large-scale features was our motivation to spectrally filter the input vector field. In the revised version we now explicitly refer back to this methodological goal in this paragraph (L. 124). We also point out an example mesoscale feature in the snapshots that might be of further interest (L. 132).

**13)** L. 127: does this relate to the two types of WCB outflow branches (cyclonic and anticyclonic outflow?) see Heitmann et al. 2024.

Thanks for pointing us to this study. We agree, there is likely a relation between the cyclonic/anticyclonic curvature of the dominant moisture transport traced by our moisture transport axes and the two types of WCB outflow branches documented by Heitmann et al. (2024). It is beyond the scope of this study to investigate the link more systematically, but we happily included a reference to Heitmann et al. (2024) as very relevant context for our observation.

**14)** L. 132 and elsewhere: "pick up" sounds a bit like slang. Can you reformulate?

We agree and replaced all occurrences by "trace" or "detect".

**15)** L. 133: "picked up by one to three of the six global detection algorithms" I don't understand what this means. Why writing "one to three" and not mentioning a clearly defined number of algorithms?

We agree this sentence was hard to follow. Different parts of our MTA is detected by different numbers of ARDTs, the strongest part by up to three algorithms, other parts generally by only one. We rephrased the sentence to make this clear.

**16)** L. 137: "expulsion of Tropical moisture" and at L. 320: "extrusions of moisture from the deep tropics". I would use an existing term such as tropical moisture export.

We agree and now use this standard term.

**17)** L. 131-140: Here you already compare your method to existing atmospheric river detection schemes, and you do this sort of evaluation again in Section 5. I would recommend to group all the comparison effort to existing methods in one method evaluation section, which I would personally prefer to have before the more phenomenological and scientific discussion about the relationship of moisture transport axes to different tropical, midlatitude and polar features.

We agree and moved all comparison to AR detection schemes to section 5 (section 4 in the revised manuscript). Section 3 in the original manuscript was intended to provide a first case study-based glimpse into what MTAs are before proceeding to the climatologies. It was thus intended to serve only as an extended illustration of the method. To make this more clear, we renumbered the (now shorter) section to be section 2.3 in the revised manuscript, a subsection of Data and Methods.

**18)** L. 176: MTA is not defined, and I preferred the written out version "moisture transport axes".

We agree and spell out the term here and in the data availability section.

**19)** L. 202: off→of

We here use "off" in the sense of "away from", so the spelling is correct.

**20)** L. 206-207: I don't understand this sentence.

Thanks for pointing this out, we rephrased to "Further, the distributions of poleward IVT for tropical and extratropical transport axes show a large degree of overlap (histogram to the side of Fig. 4d), such that a poleward IVT component is not suitable to distinguish between tropical and extratropical moisture transport."

**21)** Section 5: I find the discussion around the detection of atmospheric rivers very technical and sometimes difficult to follow. The panel d in Figures 5-9 only show a few contours from the existing atmospheric

river detection in the climatological plots and the chosen way of illustrating this intercomparison makes it difficult to compare the new method quantitatively with existing ones. If possible, I would separate the evaluation of the method based on a comparison to others from the scientifically interesting discussion of what the new method detects and what we can learn from it about atmospheric river dynamics.

The atmospheric river community focuses on the MERRA-2 reanalysis. Detections for ERA5 are thus unfortunately only available for a few AR detection schemes. We have included all available global AR detection schemes for ERA5 in our analysis.

We realise the way we explained the detection rates for the global AR detections was incomplete and thus difficult to understand. We previously presented the average detection rate across all available detection algorithms for the given composite. In the revised version, we instead show composites of the median AR detection, i.e. the frequency in which the majority of the available algorithms detect AR conditions. This method should be more intuitively meaningful, and we now explain the procedure in detail in the caption of Figure 5. Figures 1a,b of Lora et al. 2020 illustrate the difference between mean and median detections for the global AR detection climatology.

**22)** L. 262-269: I find this way of approaching the identified tropical phenomena a bit awkward. Indeed, there are more persistent features linked to the Monsoon systems but there are also many transient features such as tropical plumes (Rubin et al. 2007) or tropical moisture exports (Knippertz, 2007), often times these systems are related to Rossby wave breaking and are relevant for extreme precipitation in the subtropics (de Vries, 2021).

We agree, our introduction to tropical MTAs was too much rooted in atmospheric river-thinking. We reworked the introduction to tropical MTAs using the references pointed out by both reviewers. Many thanks for the additional context!

**23)** L. 286-287: are these barrier winds? What does the direction of transport depend on? Are these relevant questions for forecasting in these regions? And can you see and propose how one could address them by using moisture transport axes?

Given the size of the Himalayas, we would expect the moisture transport to be associated with barrier winds, yes, but we have not conducted a formal analysis of the deformation and pressure patterns to substantiate this interpretation.

The direction of moisture transport along the orography barrier does affect the isotopic composition of rainfall in Northwestern India (Joshi et al., 2023), indicating the direction of transport is important to understand the local water cycle. We are, however, not aware of forecasting challenges associated with the direction of moisture transport.

**24)** L. 295: I think here one could establish a link with the West African Monsoon (Fink et al. 2017).

Yes, this seems to be essential context. We added a reference to the West African Monsoon, but citing Sultan and Janicot (2003) and Maranan et al. (2018) instead of the suggested book chapter of Fink et al. (2017) as we could not get access to the latter.

**25)** L. 297: Yes, indeed, that is also, when most precipitation falls in the Sahara (Armon et al. 2024).

Thanks for the additional context, we added Knippertz (2007) and Armon et al. (2024) to support our statement.

**26)** Fig. 3 and others: Probably it is just my printer but the delimitation of the continents is barely (not) visible and makes it a bit difficult to orient

We agree, the shading for the continents was too pale and we darkened it in all Figures to increase their visibility.

**27)** L. 305 recycling vs. large-scale transport: A large share of moisture in cyclone precipitation is fed through the cyclone feeder airstream and originates from the cold sector of previous cyclones as well as the cyclone-anticyclone interaction zone (see, Papritz et al. 2021). Could moisture transport axes be combined with cyclone masks and tracks to study the cyclone-to-cyclone moisture hand over and multi-cyclone association of intense moisture transport in more detail? What about the temporal evolution of moisture transport axes? Can two subsequent moisture transport axes be related to each other?

It should be possible to address most of the questions using our detections. Specifically, we have in the meantime developed a tracking algorithm for jet axes that will be applicable also for MTAs. So yes, two subsequent MTAs can already be related. We have not combined MTAs and cyclone masks beyond some preliminary tests, but we would expect this to be rather straightforward.

The main obstacle for answering the reviewer's questions will likely be the attribution of moisture transport to MTAs. After all, MTAs only mark the line of maximum transport, which is not enough to calculate a transport budget. However, some of the ideas of the precipitation attribution work by Konstali et al. (2024a,b) might be suitable to fill this methodological gap.

**28)** L. 307-311: Yes, the tracing of the moisture filaments further into the subtropics, continents and/or polar regions is a very nice characteristic of the new method but here I think the Eulerian vs. Lagrangian aspect should be mentioned and discussed as a caveat resp. as a possible outlook: this of course does not mean that you suggest that moisture transport in the atmosphere is generally occurring over longer distances or timescales. A combination with a trajectory-based diagnostic would be required for investigating the moisture cycling aspect along these moisture filaments.

We agree and introduced a new paragraph in which we discuss the advantanges and disadvantages of our approach relative to atmospheric river detections and Lagrangian air stream definitions (L. 309-319).

**29)** L. 311: yes, and I think this is really exciting because it allows to study the substructure of atmospheric rivers in more detail, in particular the relevance of moisture recycling through precipitation evaporation and the importance of cold pool-like circulations within the complex cloud-systems.

Very good to read, thanks for the positive feedback!

**30)** L. 320: "extrusions of moisture from the deep tropics"→tropical moisture exports? Or is the feature you describe something meteorologically different?

No, we meant to the same feature that Knippertz and Wernli (2010) refer to as tropical moisture exports, and now use this more standard term.

**31)** L. 324-331: yes, exactly, very nice and convincing concluding paragraph, that's the framing I would also strongly encourage to adopt for the introduction.

Very good to read, thanks! We revised the introduction following the same framing.

**32)** I miss a serious discussion of the caveats of the method in the conclusions and an outlook about which scientific questions could be addressed with this new valuable detection scheme.

We agree, and added a paragraph on caveats (L. 309-319) relative to both Lagrangian air stream definitions and atmospheric rivers.

**33)** L. 339: TWe→We

Thanks for pointing out the typo, we corrected it.

**34)** L. 333: I commend the authors on their plan to make their climatology of moisture transport axes publicly available. MTAs have not been introduced as an abbreviation→write it out?

Thanks for pointing out the undefined abbreviation, we now spell out the name.

**Reviewer 2**

**Major concern, part I** The manuscript represents objective global climatology of moisture transport axes. The paper is well-written and the figures are well designed. The method effectively captures moisture transport outside the tropics, including high latitudes.

My major concern is the moisture transport axes (MTAs) within the deep tropics. Specifically, Fig. 3 shows a high frequency of MTAs in the ITCZ region, which I interpret as moisture transport along the ITCZ from east to west. As far as I understand, the ITCZ signifies moisture convergence driven by surface trade winds, typically not organized into distinct filaments. In this instance, the method identifies convergence into the ITCZ as a peak in transport and creates an impression of moisture transport along the axis, not into the axis.

In Supplements, the authors show an alternative method based on normalised water vapour transport. This method gives similar results in extratropics but does not detect ITCZ as an MTA; therefore, in my view, it is better suited for the identification of MTAs. Alternatively, separating MTAs into two types, e.g., dominated by transport along the axis vs transport into the axis or dominated by advection vs water content, might help avoid confusion.

We agree with the reviewer that we detect a relatively large number of MTAs in the deep tropics which might not be desirable for all applications. However, we want to point out that the detection rate in the

[Figure]

Figure 1: As Fig. 3 in the manuscript, but for January-February-March and focusing on Australia.

deep tropics is still only 20-30% in the Tropical Pacific and less in the Tropical Atlantic. In contrast to moisture convergence, MTAs in the deep tropics are thus far from a permanent feature. For example, the broad region of moisture transport across the Tropical Atlantic in Fig. 2g,h is not detected as an MTA because it lacks a clear maximum.

We have had long discussions both internally and with the reviewers of a previous submission of this work at JGR-A on how to include the normalised detections. In the end we settled on keeping the analysis as a supplement, because, as the reviewer points out, this step might be useful for some tropical applications. At the same time it is an unnecessary complication involving unnecessary choices without benefits for the mid-latitudes and potentially more spurious detections at high latitudes. We feel we have thus achieved a reasonable compromise.

Finally, thanks for the suggestion to differentiate between wet and windy MTAs. We like the suggestion and might follow up on the idea in a subsequent publication. For the present work, we would like to point the reviewer to the two-dimensional histograms in Fig 4, where panel (a) indeed shows a bimodal distribution of TCWV with the two peaks representing tropical and extratropical MTAs, respectively. The bi-modality is however clearer in latitude (histogram on top of Fig 4a); for the sole purpose of distinguishing tropical from extratropical MTAs, a simple latitude threshold would thus yield a more precise separation than a separation of wet and windy rivers through a threshold in TCWV. Following one of your suggestions below, we now separate tropical and extratropical MTAs in the histograms on the sides of Fig. 4.

**Major comment, part II** Furthermore, the authors assert that the method is well-suited for monsoons based on good performance in India and West Africa. However, it appears that, e.g., the Australian monsoon, which brings moisture toward the north of the Australian continent in January-March/April, is not explicitly represented. It may be that moisture transport in this region is not organised in filaments. If this is the case, discussing this nuance in the manuscript could provide a more comprehensive view of the method's applicability across different monsoon systems.

Thanks for pointing out this gap in our analysis, we include the MTA climatology for January-March around Australia as Figure 1 in this document. During the monsoon season, MTAs are detected for about 10-25% of the time steps over large parts Northern Australia. In contrast, detection rates in this region during southern winter (DJF) are generally below 2.5% (Fig. 3c in the manuscript). We thus do detect features of the Australian monsoon, but admittedly considerably less prominent than its Indian counterpart. We now mention the Australian monsoon explicitly (L274-279).

**Major comment, part III** Finally, it would be good to cite the recently published paper (Konstali et al. 2024) that utilises MTAs described in this study. Even though Konstali et al. (2024) cite the pre-print of the manuscript under review submitted to another journal, it would be worthwhile adding a citation of Konstali et al. (2024) in this manuscript highlighting how MTAs contribute to rainfall in different parts

of the globe. Following the approach outlined in Konstali et al. (2024), it may be valuable to investigate how frequently MTAs are linked with cyclones and/or fronts, especially in light of findings by Spensberger and Spengler (2018), which demonstrated that heat and moisture transport can be effectively utilised for classification of fronts. Integrating this analysis into Section 5 of the manuscript could significantly enhance the discussion.

We very much agree and included Konstali et al. (2024b) in the revised introduction. We also very much like the idea to link the MTA detection to the front classification suggested by Spensberger and Sprenger (2018). We now mention the front classification in our discussion of mid-latitude MTAs (L. 174) and we might follow up on that idea in more detailed future work. To some degree we do already take this into account in Konstali et al. (2024a,b) by considering MTAs in combination with the Spensberger and Sprenger (2018) frontal volumes and both fronts and MTAs in isolation.

**Specific comments**

**l. 38-42** I would disagree with the statement that 'it is unclear to what extent the concept [of AR] can and should be extended to the subtropics.' Subtropical latitudes are indeed affected by extratropical weather systems, as you suggested in that paragraph, more often in winter, but also warmer months. Catto et al (2015) show that the highest percentage of cold fronts associated with WCB are found between 20-30degS since the majority of WCBs in their dataset are found equatorward of 40degS. Some papers use the term AR in relation to Australian rainfall (e.g., Rauber et al 2022 (10.1029/2020JD032513), Reid et al. 2020, 2021, 2022). I would argue that subtropics are regions of active ARs as high moisture sources are nearby (Gimeno et al 2021). I think Fig4b of the submitted manuscript supports the idea that moisture is actively transported in the subtropical regions (15-25 deg), depite a relatively low total column water (Fig. 4a). Even though the distribution of axes frequencies at the top of Fig.4 shows a dip in the subtropics, these MTAs are important and might be responsible for extreme rainfall in subtropics and low midlatitudes.

Thanks for bringing this up and for the additional references. We were aware of some work on subtropical ARs, but not of the extent to which the AR concept has already been transferred to the low latitudes. In the light of comments from both reviewers, we reworked our framing of tropical MTAs in the introduction and removed the first paragraph of the section on tropical MTAs (section 6 in the revised manuscript). We found the remainder of the section to work well without further introduction.

   We agree with the reviewer on the meteorological significance of subtropical moisture transports. Indeed, Fig 4b shows that typical IVT values do not differ substantially between the tropics, subtropics and extratropics with the most frequent value around 300-350 kg/(s m). At the same time, the gap at subtropical latitudes in Fig 4a does not imply low TCWV, but only that MTAs are relatively infrequent at these latitudes. The darkest shading in Fig 4a for 15-20° latitude is around 40-50 kg/m$^2$.

**l.93** A single threshold based on the global average is potentially biased towards low latitudes with higher moisture content and, therefore, contradicts the idea of avoiding thresholds that you postulated at the very beginning.

We disagree. The single threshold is for a metric taking into account the sharpness of the IVT maximum ($\frac{\partial \sigma_{IVT}}{\partial n}$) and the absolute IVT magnitude $|IVT|$. By combining these two components into one threshold, we do both detect relatively broad maxima at high absolute IVT as well as narrow and pronounced IVT maxima nearly irrespective of the absolute IVT magnitude. Removing the contribution from $|IVT|$, as reviewer 1 suggests, would have the following two (generally undesirable) consequences: (a) IVT-maxima of intermediate sharpness will be detected also along very weak moisture filaments, (b) somewhat diffuse maxima along very strong moisture filaments might no longer be detected. In addition to detecting a number of moisture features that are irrelevant for weather, we might thus also no longer detect some unambiguous atmospheric river cases, for example when a secondary cyclogenesis along a cold front locally distorts the structure of the (otherwise strong) moisture transport.

**L.130** In fig. 2d, is the transport away from a cyclone associated with a warm front? It seems to be too far from the cyclone centre to be a part of a warm front but maybe it still is.

Yes, that's exactly it. It likely also represents the anticyclonically curving part of the WCB outflow documented in Heitmann et al. (2024; reference through reviewer 1). At the same time, we have the same hunch as the reviewer and question to what extent the warm sector and cyclone core of this cyclone are still dynamically related. As potentially interesting context, Spensberger and Schemm (2020) consider

the same question in more detail for a cyclone making landfall on the Norwegian West Coast. In this case study, orography separated the cyclone core from its warm sector, apparently without affecting the life of the cyclone core too much.

**l.135** The moisture transport around a TC does not represent the actual moisture transport which has not been identified as AR by any of the algorithms in ARTMIP-ERA5. Since TCs are relatively rare, their contributions to climatology are small. However, is there a way to remove or improve those axes?

Yes, there are schemes that explicitly remove the moisture transport around TCs. For example, Tempest-Extremes v2.1 explicitly disregards moisture transport within 8° of detected TC cores (Ullrich et al. 2021). If so desired, it would be straightforward to filter our MTAs using a similar criterion. The choice of whether or not to exclude MTAs close to TCs will depend on the application, and we thus do not introduce a TC-filter in the present work.

**l. 147** (raised in my general comments) To what extent does ITCZ advect moisture in the zonal direction? My understanding of ITCZ is that it represents the convergence of moisture advected from subtropics. In my view, this advection mostly happens not in filaments. Fig.4 (c,d), tropical moisture axes advect moisture eastward and mainly equatorward.

We think our tropical detection rates in Fig. 3 and tropical snapshot in Fig. 2g,h are generally conform with these expectations. Further, Figs. 4c,d show that the moisture transport is clearly more westward than equatorward for tropical MTAs. For more details we refer to our response to the general comment.

**l. 200** The peak around 60S in Fig. 4c is weak but can be defined. Can eastward flux at 60 deg lat be associated more with eateries around Antarctica than circulation around the cyclone centre, which usually happens over short distances? I am not sure how this can be objectively measured other than separating the moisture transport into transient and stationary components (which is an interesting avenue, perhaps).

Good point, we included the suggestion as a second potential explanation for the westward transport around 60° latitude. The number of points comprising this peak is however at least an order of magnitude smaller than the mid-latitude peak in eastward transport, implying that on average less than a tenth of the MTA is pointing westward poleward of a cyclone. Based on our synoptic experience detecting MTAs in operational ECWMF forecasts, this magnitude does not seem unreasonable.

**l.205** I'd recommend showing distributions for tropical and extratropical MTAs separately as they may represent different processes. Also, as far as I understand, this plot shows moisture transport only along the axes, it does not tell you how much moisture is advected within atmospheric rivers as a whole. Therefore, the distribution of moisture transport within atmospheric rivers might be more poleward.

Good idea, thanks for the suggestion! We have in the revised manuscript split the histograms on the sides/top of Figure 4 into tropical and extratropical MTAs.
   The reviewer is correct that the histograms only show the moisture transport at the location of the detected MTA, not for an atmospheric river-like object as a whole. However, the consequence of this discrepancy is generally in the opposite direction as the reviewer suspects. For eastward propagating moisture filaments the transport direction will generally be more zonal than the orientation of the filament, and thus the orientation of the detected MTA. The snapshots in Figs. 2a,e showcase this nicely in that most IVT vectors are veered anticyclonically relative to the moisture filaments following mid-latitude fronts.

**l.218** You mention that high latitude ARs/moisture transport is similar to AR in the midlatitudes. In the two Antarctic cases shown in the paper, AR/MTA stretches from the subtropics to the Antarctic coastline. What do you mean by saying that those high-latitude events are 'similar' to their 'midlatitude counterparts'? I read that they are similar but still different (maybe that's not what you meant). In my view, they are created by the same process, i.e., moisture advection associated with a cyclone/frontal circulation that must be the leading cause of ARs formation.

We agree with the reviewer that from a fundamental fluid dynamics-perspective the dynamical origin of MTAs will be the same in the mid- and high latitudes: strain-driven frontogenesis. However, from a weather perspective there are still conceptual differences between the regions. In the mid-latitudes, the strain will predominantly be associated with cyclones and their fronts, whereas in polar regions the palette of weather systems is a bit more diverse. In addition to cyclones with mid-latitude structure, there are also cold-air outbreak boundaries, polar lows, and tropopause polar vortices that can be associated

with frontogenetic strain in the flow field leading to polar MTAs. In the light of this larger conceptual diversity of polar weather, our formulation seems appropriate.

**l.252** My reading of Papritz et al is they explore moisture transport into the Arctic. Can you mention their Figure that shows the extrusion of the warm moist air from the polar region? Also, the statement that the transport around the cyclone centre advects air that is still relatively warm should be supported by temperature analysis. As a side comment, it would be interesting if a similar approach could be applied for heat advection.

Papritz and Dunn-Sigouin do never show the export directly, but only what they call net and total fluxes (throughout all figures). Yet, by systematically considering the differences between net and total moisture imports, they show that not all moisture imported into the Arctic stays there.

We do show anecdotal evidence for our interpretation in Fig. 2c, where air with TCWV well above $20 \, \text{kg/m}^2$ recurves around the cyclone's occlusion point and is transported equatorward again. For air to hold that amount of moisture, it must be rather warm by subpolar standards. In the light of this, we don't think it is necessary to back up this side remark by a dedicated analysis.

**l.264** Subtropical climate is very variable. In winter, extratropical weather systems are often observed in subtropics, especially in the area below the subtropical jet that increases baroclinicity. In warmer months, extratropical circulation shifts further poleward but fronts are still frequently intruding subtropics. Either frontal circulation or cutoff lows can create a strong moisture advection in the subtropics, particularly in late summer-early autumn. ARs are important for rainfall in South Africa (e.g., Blamey et al 2018), Australia (e.g., Reid et al 2022), subtropical South America (Reis et al. 2022), the Middle East and North Africa (Massoud et al. 2020), Southeast China (Xu et al. 2020 https://doi.org/10.1071/ES19027). The subtropical climate varies a lot from one region to another but there are many subtropical areas for which moisture transport in the form of ARs is critically important as they create extreme rainfall events.

We agree that our introduction to subtropical and tropical MTAs was misleading. We removed the introductory paragraph and found the remainder of the section to work well without replacedment. Further, we extended the discussion using the comments and references provided by both reviewers. Many thanks for suggesting all the additional relevant context!

**l. 292** Can you explain why rainfall associated with this moisture transport remained weak given a close proximity to the ITCZ? (Though it could have been strong in a relative sense in this arid region)

As the reviewer points out, the MTAs in question occur over a very arid region rather than the moist ITCZ. In this region it will generally be even warmer than it is moist, and time-mean subsidence further suppresses convection, condensation and rain formation.

**l.313** In Fig. 4c, the occurrence of poleward moisture transport in high latitudes(>60deg) is higher than the equatorward transport, suggesting that more moisture is advected into the polar regions than outside.

We agree, and that is also what we write: "Moisture transport axes thus highlight events with pronounced moisture import into polar regions."

**l.321** The moisture transport 'along the straits of the Maritime continent', also mentioned earlier in the manuscript, is interesting. Looking at seasonal rainfall (e.g., Fig 1 in Bukowski et al. 2017, 10.5194/acp-17-4611-2017), I cannot see why the moisture transport axis lies exactly between islands. If it is not an artefact, can it be explained?

Thanks for pointing us to the study of Bukowski et al. 2017. We think the focus of our detections on the straits is not an artefact. Many islands of the maritime content feature high orography, and thus block considerable parts of the moisture transport, which is strongest in the lower troposphere. The straits will thus generally feature a maximum in moisture transport relative to the neighbouring islands, which is detected as an MTA.

**Supplement, l. 14** Following Wille et al. and Gorodetskaya et al., I think, moisture transport into high latitudes is very important for polar regions. You say that a lower threshold leads to spurious detections over Antarctica. Did you check if axes that would be identified with a lower threshold were not associated with precipitation over Antarctica and, therefore, could be suspected as spurious?

We did not check this, in parts because we were unsure how much we can trust ERA5 precipitation over the Antarctic Ice Sheet. Our judgement that many of these MTAs likely are spurious is instead based on their frequency of occurrence of our normalised MTAs in the polar regions, which by far exceeded the

AR detection rates of, for example, Wille et al. (2021). We now point this reasoning out explicitly in the revised supplement.

**Technical comments**

**Fig. 1** Please define the blue shading.

Thanks for pointing out the typo/omission. The blue shading shows IVT.

**Fig. 3** I barely see continents, can you please make them more visible?

Yes, thanks for pointing out the problem. We darkened the continents in all Figures.

**Fig. 5,6,7,9** Could you please mark the location of the target area for moisture axes with a symbol? Are Fig.5 (c,d) and subsequent plots shown for a particular season?

Thanks for the suggestion. We have experimented with adding markers and in the end decided against. The target location is already marked well by the maximum in the MTA detection frequency, and generally also by a local maximum in the IWV. Additional markers for the same location made in our experiments the plot only harder to read. Instead we added a note to the caption of Figure 5 on how to identify the target location in the plot. And no, the composites are defined year-round, and thus follow the seasonality of MTAs in the respective location. Thanks for pointing out that missing information.

**References**

Berry, G., Reeder, M. J., and Jakob, C.: A global climatology of atmospheric fronts, Geophysical Research Letters, 38, L04 809, https://doi.org/10.1029/2010GL046451, 2011.

Jenkner, J., Sprenger, M., Schwenk, I., Schwierz, C., Dierer, S., and Leuenberger, D.: Detection and climatology of fronts in a high-resolution model reanalysis over the Alps, Meteorological Applications, 17, 1–18, https://doi.org/10.1002/met.142, 2010.

Joshi, S. K., Kumar, S., Sinha, R., Rai, S. P., Khobragade, S., and Rao, M. S.: Identifying moisture transport pathways for north-west India, Geological Journal, 58, 4428–4440, https://doi.org/10.1002/gj.4759, 2023.

Knippertz, P. and Wernli, H.: A Lagrangian Climatology of Tropical Moisture Exports to the Northern Hemispheric Extratropics, Journal of Climate, 23, 987–1003, https://doi.org/10.1175/2009JCLI3333.1, 2010.

Konstali, K., Spengler, T., Spensberger, C., and Sorteberg, A.: Linking Future Precipitation Changes to Weather Features in CESM2-LE, Journal of Geophysical Research: Atmospheres, 129, e2024JD041 190, https://doi.org/10.1029/2024JD041190, 2024a.

Konstali, K., Spensberger, C., Spengler, T., and Sorteberg, A.: Global Attribution of Precipitation to Weather Features, Journal of Climate, 37, 1181–1196, https://doi.org/10.1175/JCLI-D-23-0293.1, 2024b.

Maranan, M., Fink, A. H., and Knippertz, P.: Rainfall types over southern West Africa: Objective identification, climatology and synoptic environment, Quarterly Journal of the Royal Meteorological Society, 144, 1628–1648, https://doi.org/10.1002/qj.3345, 2018.

Schemm, S., Rudeva, I., and Simmonds, I.: Extratropical fronts in the lower troposphere–global perspectives obtained from two automated methods, Quarterly Journal of the Royal Meteorological Society, 141, 1686–1698, https://doi.org/10.1002/qj.2471, 2015.

Spensberger, C. and Schemm, S.: Front-orography interactions during landfall of the 1992 New Year's Day Storm, Weather and Climate Dynamics, 1, 175–189, https://doi.org/10.5194/wcd-1-175-2020, 2020.

Spensberger, C. and Spengler, T.: Feature-Based Jet Variability in the Upper Troposphere, Journal of Climate, 33, 6849–6871, https://doi.org/10.1175/JCLI-D-19-0715.1, 2020.

Sultan, B. and Janicot, S.: The West African Monsoon Dynamics. Part II: The "Preonset" and "Onset" of the Summer Monsoon, Journal of Climate, 16, 3407–3427, https://doi.org/10.1175/1520-0442(2003)016⟨3407:TWAMDP⟩2.0.CO;2, 2003.

---

## Author Response (AR2)

**Response to reviewers – "Moisture transport axes: a unifying definition for monsoon air streams, atmospheric rivers, and warm moist intrusions"**

C. Spensberger, K. Konstali & T. Spengler

16 December 2024

We thank Franziska Aemisegger and the anonymous reviewer for reviewing the revised manuscript. We are happy to read that we were generally able to satisfy the reviewers and thank for pointing out some remaining issues. Our point-by-point response appears below in blue.

**Reviewer 1 (F. Aemisegger)**

**L 70** can you find a smoother way to lead over from the literature review to the data and methods section. The end of the intro is a bit abrupt.

Thanks for pointing this out. We agree that the transition was a bit abrupt. We have considered several options to smooth the transition by appending a sentence or two to the end of the introduction, like a final summing up and concluding of the literature review or a table-of-contents sentence, but in the end decided against these options. We feel that a conclusion would merely constitute a repetition of what we pointed out before and we generally find table-of-content sentences/paragraphs to be redundant.

To still smooth the transition somewhat, we changed the beginning of the data subsection, which now reads "We detect moisture transport axes in 3-hourly ERA5 reanalysis [...]". By taking up the phrase "moisture transport axes" from the previous sentence, we now provide a bit more continuity and a smoother transition.

**code availability** the jet axis detection algorithm is available in the dynlib, but adaptations are necessary to use it for moisture transport axes. Could these changes be documented in the supplementary, or the adapted code for moisture transport axes be made available?

Thanks for pointing out this missing bit of information. In fact, no adaptations of the algorithms are required. We now state this explicitly in the revised code availability statement: "The jet detection algorithm is used as published, no adaptations to the algorithm are required beyond setting the appropriate threshold."

**Reviewer 2**

**General remark** I am still not fully convinced that a single global threshold is the best approach. I agree with the arguments in the response letter that removing —IVT— could result in peaks in the gradient field that are not associated with significant moisture content but would still be identified as moisture transport axes. I believe that if the Kivt parameter were to vary with latitude, following, e.g., the smoothed IVT zonal profile, it would provide a more accurate representation of moisture transport in mid- and high-latitudes. (lines numbers in the manuscript with tracked changes)

We realise that this is a choice where we will not be able to avoid some disagreement among readers and reviewers, irrespective of our choice. We have followed the performance of the algorithm for more than two years now in the weekly chart discussions at our department and we genuinely believe that the variant we propose in the manuscript is the most generically useful one (in the extratropics, see discussion in the supplement.)

Having said that, we invite the reviewer to test his/her ideas and maybe suggest an improved definition for the feature. With the acceptance of the manuscript we will also publish the detections described in the

supplement, where we normalise the magnitude of the input IVT by the time and zonal average TCWV. This approach is equivalent to a latitudinally-varying $K_{ivt}$-threshold, so this might be an interesting dataset for the reviewer to consider.

**l.16** I believe the last comma is not needed, i.e., it should read 'mid-latitudes and polar regions'

True, it is a serial comma and thus optional—but it is a valid choice to have it there. We consistently use the serial comma in the manuscript (including the title) and thus opt to keep it.

**l.39** 'the concept of atmospheric rivers'

We agree with the suggested edit and adapted accordingly.

**l.202, 271** I would recommend not using the word 'detections' in those captions. I like your old subtitle of Section 5 better (Moisture transport axes in polar regions/midlatitudes and their relation to...)

We agree, the word "detections" makes the titles appear more technical than appropriate. We thus replace "detections" by "moisture transport axes" in the titles of sections 4-6.

**l.219-222** This part should be made clearer and perhaps shorter, e.g., 'Not all fronts, however, are associated with moisture transport axes (e.g., warm fronts south of South Africa in Fig. 2c,d). Thus, the co-occurrence of fronts and moisture transport axes may define frontal moist baroclinicity - a metric that combines relative humidity and the isentropic slope, effectively describing fronts (S&S2018)."

Thanks for suggesting a more elegant phrasing for what we wanted to express there. We agree with the suggestion and changed the wording accordingly.

**l.371-374** 'attribution' of what? also, the word attribution is used 3 times in one sentence

This sentence is meant generically, i.e., the attribution of any given variable. We rephrased to make this clearer ("Compared to atmospheric rivers, moisture transport axes offer a more structure-based definition, at the cost of making it less straightforward to relate moisture transport axes to, for example, poleward vapour transport or precipitation.") and adapted the following sentence to avoid the repetition of "attribution".

**l.399** 'our approach allows (us) to unify atmospheric rivers'. Please bring 'us' back to the phrase.

Fine for us, we re-added "us".